# A central role for the retrosplenial cortex in de novo environmental learning

Stephen D Auger, Peter Zeidman, Eleanor A Maguire*

Wellcome Trust Centre for Neuroimaging, Institute of Neurology, University College London, London, United Kingdom

**Abstract** With experience we become accustomed to the types of environments that we normally encounter as we navigate in the world. But how does this fundamental knowledge develop in the first place and what brain regions are involved? To examine de novo environmental learning, we created an 'alien' virtual reality world populated with landmarks of which participants had no prior experience. They learned about this environment by moving within it during functional MRI (fMRI) scanning while we tracked their evolving knowledge. Retrosplenial cortex (RSC) played a central and highly selective role by representing only the most stable, permanent features in this world. Subsequently, increased coupling was noted between RSC and hippocampus, with hippocampus then expressing knowledge of permanent landmark locations and overall environmental layout. Studying how environmental representations emerge from scratch provided a new window into the information processing underpinning the brain's navigation system, highlighting the key influence of the RSC.

## Introduction

We continually encounter new environments and to operate effectively within them we must be able to form dependable representations of these surroundings. Numerous brain areas, including the hippocampus (*O'Keefe and Nadel, 1978*; *Rosenbaum et al., 2004*; *Spiers and Maguire, 2006*; *Bird and Burgess, 2008*; *Manns and Eichenbaum, 2009*; *Howard and Eichenbaum, 2014*), entorhinal (*Hafting et al., 2005*; *Sargolini et al., 2006*; *Doeller et al., 2010*; *Chadwick et al., 2015*), parahippocampal (PHC; *Janzen and van Turennout, 2004*; *Kravitz et al., 2011*; *Mullally and Maguire, 2011*; *Epstein and Vass, 2014*), retrosplenial (RSC; *Chen et al., 1994*; *Cho and Sharp, 2001*; *Vann et al., 2009*; *Auger et al., 2012*; *Auger and Maguire, 2013*) and posterior parietal (*DiMattia and Kesner, 1988*; *Husain and Nachev, 2007*) cortices have been implicated in representing space and facilitating navigation therein, with generally concordant findings across species. We still lack, however, a precise account of how an environmental representation evolves de novo and the roles played by space-sensitive brain regions in this process. It is surprising that such a gap in our knowledge exists given that learning new environments is ubiquitous and relevant for independent living and even survival. Moreover, examining the genesis of environmental representations would seem an ideal means of leveraging our understanding of the specific roles played by relevant brain areas, when they come online, what they respond to, and how and when regions interact with each other to support the emerging representation.

In recent years, there has been a move to study how neuronal responses in the medial temporal lobes develop in rodent pups when they interact with the world for the first time (*Langston et al., 2010*; *Wills et al., 2010*). This exciting research is only just starting to provide clues about when specific brain areas become functional and in response to what, leaving much still to learn about the neural development of spatial cognition and memory (*Mullally and Maguire, 2014*; *Wills and Cacucci, 2014*). In adult humans it is possible to study how multiple brain areas are engaged during

*For correspondence: e.maguire@ucl.ac.uk

**Competing interests:** The authors declare that no competing interests exist.

**eLife digest** Throughout our lives, we encounter novel environments that we must learn to find our way around, from a new office to a new city. Studies of brain activity in humans and rodents have revealed that many brain regions are involved in navigation, most notably the hippocampus. However, these experiments have typically involved humans navigating around environments filled with familiar objects and landmarks, and therefore tell us relatively little about how the brain builds up a map of a completely new environment in the first place.

To address this issue, Auger et al. scanned the brains of healthy human volunteers as they experienced an 'alien' virtual reality world called 'Fog World', so-named because of the dense fog used to precisely control what the volunteers could see. In contrast to previous virtual reality environments, which have contained houses, shops and other recognisable objects, Fog World contains only abstract landmarks that bear little resemblance to anything in the real world. The volunteers watched videos that simulated journeys through Fog World with the goal of learning the layout of the environment so that they could navigate within it. Of note, half of the landmarks in Fog World remained in fixed positions on all learning trials, while the other half changed location from one trial to the next.

After each block of trials, the volunteers were shown single landmarks—some from Fog World and others not—while their brains were scanned. A region called the retrosplenial cortex showed increasing activity that closely tracked the volunteers' growing knowledge of which landmarks had fixed, permanent locations in Fog World. In later trials towards the end of the learning period, the hippocampus also became active, and at this time communication between the retrosplenial cortex and hippocampus was also heightened. By the end of learning, the hippocampal activity was related to the volunteers' knowledge of the locations of the permanent landmarks across Fog World.

As well as revealing that the retrosplenial cortex may be essential for processing permanent landmarks, the work of Auger et al. shows how the hippocampus and retrosplenial cortex could work together to map new environments. These findings might also help us to better understand why some healthy individuals are bad navigators, and why disorientation is a common early symptom in neurodegenerative disorders such as Alzheimer's disease, where the retrosplenial cortex is often one of the first brain regions to become damaged.

navigation in large-scale environments by having subjects navigate in virtual reality (VR) while being scanned using techniques such as functional MRI (fMRI). In the majority of these experiments subjects become familiarised with an environment before scanning and then typically perform tasks during scanning based on the environmental representation they formed pre-scan (e.g., *Janzen and van Turennout, 2004*; *Spiers and Maguire, 2006*). A smaller number of VR scanning studies have focused on the acquisition phase of environmental knowledge (e.g., *Wolbers et al., 2004*; *Wolbers and Buchel, 2005*; *Iaria et al., 2007*; *Baumann et al., 2010*). However, as far as we are aware, in every case the environments contained landmarks and structures that were readily recognisable and nameable (e.g., shops and houses). Thus, the neural substrates of learning about an environment from scratch, with no prior experience of, or knowledge about, key elements within it, has never been examined in humans.

How children and adult humans learn about new environments has been studied extensively in cognitive and environmental psychology. Prominent features in an environment, namely landmarks, have been posited to play a fundamental role (*Tolman, 1948*; *Lynch, 1960*; *Siegel and White, 1975*; *Downs and Stea, 1977*; *Golledge, 1991*; *Lew, 2011*). It has been further suggested that encoding landmarks facilitates the development of route knowledge, and learning how routes relate to each other then provides the navigator with a 'survey' representation of an environment (*Siegel and White, 1975*; *Epstein and Vass, 2014*). Despite landmarks being heavily implicated in many neuroscientific experiments in animals and humans over the decades, the effect of landmark properties on navigation has only recently been studied. *Auger et al. (2012)*; (see also *Mullally and Maguire, 2011*; *Konkle and Oliva, 2012*; *Auger and Maguire, 2013*) using fMRI found that PHC responded to visuospatial features of landmarks. By contrast RSC was engaged only when subjects viewed landmarks that had a permanent location and never moved (a finding that has since been replicated—*Marchette et al., 2014*;

*Troiani et al., 2014*). This suggests that a function of the RSC may be to code for permanent landmarks on which a stable spatial representation of an environment can then be built. But, as with the work mentioned above, the landmarks examined in these studies were of a type already familiar to subjects.

Key questions therefore remain. How does knowledge of landmark features, including permanence, but also properties such as size and visual salience, evolve de novo during environmental learning where no prior semantic knowledge exists about landmarks? What brain areas support this learning, at what point do they come online, and to what aspects of the environment do they respond? Furthermore, how might this information be used in building an overall environmental representation? We addressed these questions in the current study.

To do so we developed a new VR environment that subjects learnt during repeated exposures while undergoing fMRI scanning, and their accruing knowledge was tested during and after scanning. The environment was populated by entirely novel, 'alien' landmarks (*Figure 1A*) about which subjects had no pre-conceived ideas. These were located along different paths (*Figure 1B*; 'Materials and methods'). On each path, half the landmarks were permanent, each remaining fixed in a single place, while the rest were transient and changed location on every exposure. The landmarks were developed and characterised in an initial experiment with separate subjects (detailed in the 'Materials and methods') ensuring that the permanent and transient landmark groups were matched in terms of visual salience, how well they could be remembered, as well as other features.

Before scanning, subjects were instructed to learn the layout of the environment and told that they would be tested in a variety of ways after scanning without the specific nature of those tasks being revealed. They were informed that some of the landmarks would always remain in the same location whereas others would appear in a different place every time they saw them. The world contained five different coloured intersecting straight paths (yellow, red, grey, blue and green; *Figure 1C*). Each path had 12 landmarks (six permanent, six transient) evenly distributed alongside it (*Figure 1B*). While

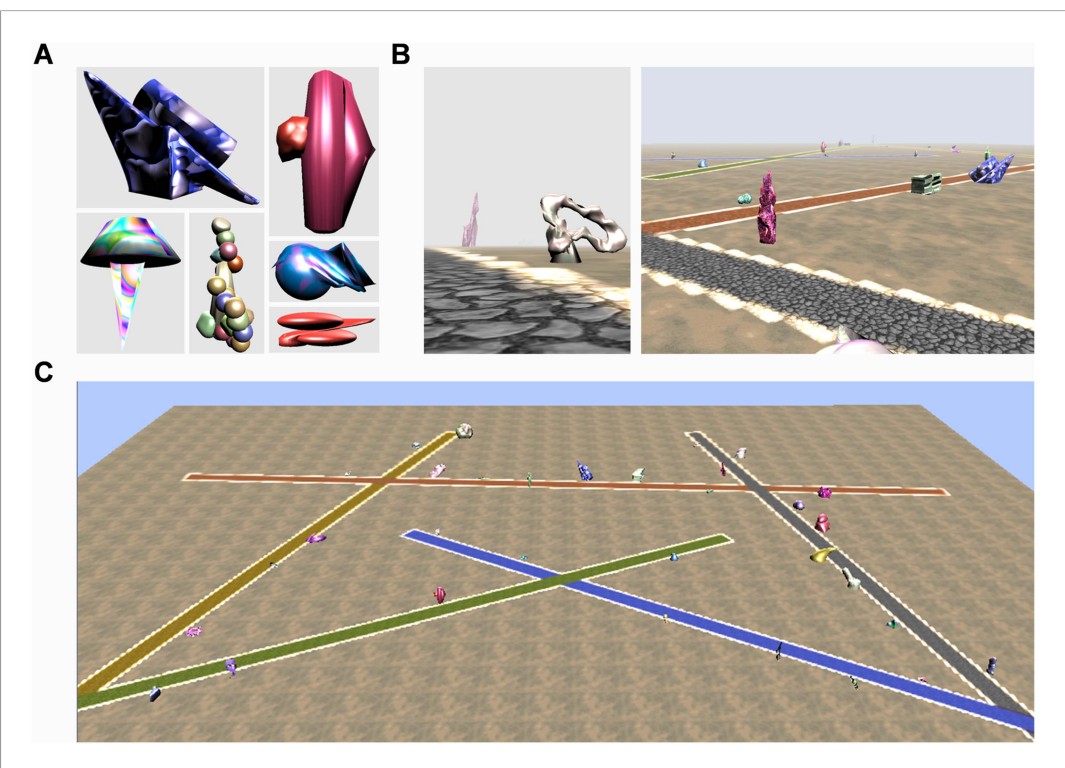

**Figure 1**. The virtual reality environment 'Fog World'. (**A**) Examples of the 'alien' landmarks. (**B**) Landmarks positioned within the virtual world. (**C**) An overhead perspective of the environment showing the five different coloured, intersecting paths—note this aerial view was never seen by participants during learning.

undergoing fMRI scanning, subjects learned the layout of the environment and its landmarks by viewing first person perspective videos travelling along each of the five paths, one at a time. Each trial consisted of a single journey along one of the paths and at the end of a video subjects were immediately shown the next learning trial on a different path. In these videos, the environment was covered in a shroud of fog to restrict the field of view thus ensuring we had complete control over the exposure subjects had to each landmark (*Figure 2A*)—hence we refer to the environment as 'Fog World' (see *Video 1*).

When all five paths had been travelled once, there came a questioning period to gauge how much information subjects had learned by that point in the experiment (*Figure 2B*). In these questioning periods, participants were first shown an image of a single landmark displayed, in isolation, on a plain grey background for 2 s. They were then asked whether or not they remembered the landmark from the environment ('*Have you seen this item in the environment?*', Yes/No). If they remembered seeing it, they were then asked about its permanence ('*How many locations in the environment have you seen it in?*', Only 1/More than 1), before being questioned about another landmark. Within each questioning period, subjects were asked about 13 landmarks: five permanent, five transient and three previously unseen. The combination of a 13 landmark questioning period and videos of the five different paths preceding it are referred to as a learning 'sweep'. There were three such sweeps in each scanning run (or quarter) and four runs, so in total 12 learning sweeps. Once out of the scanner after learning had concluded, subjects' knowledge of Fog World landmarks (recognition memory or 'memorableness', permanence, visual salience and size—see 'Materials and methods'), as well as their ability to volitionally navigate within Fog World were assessed.

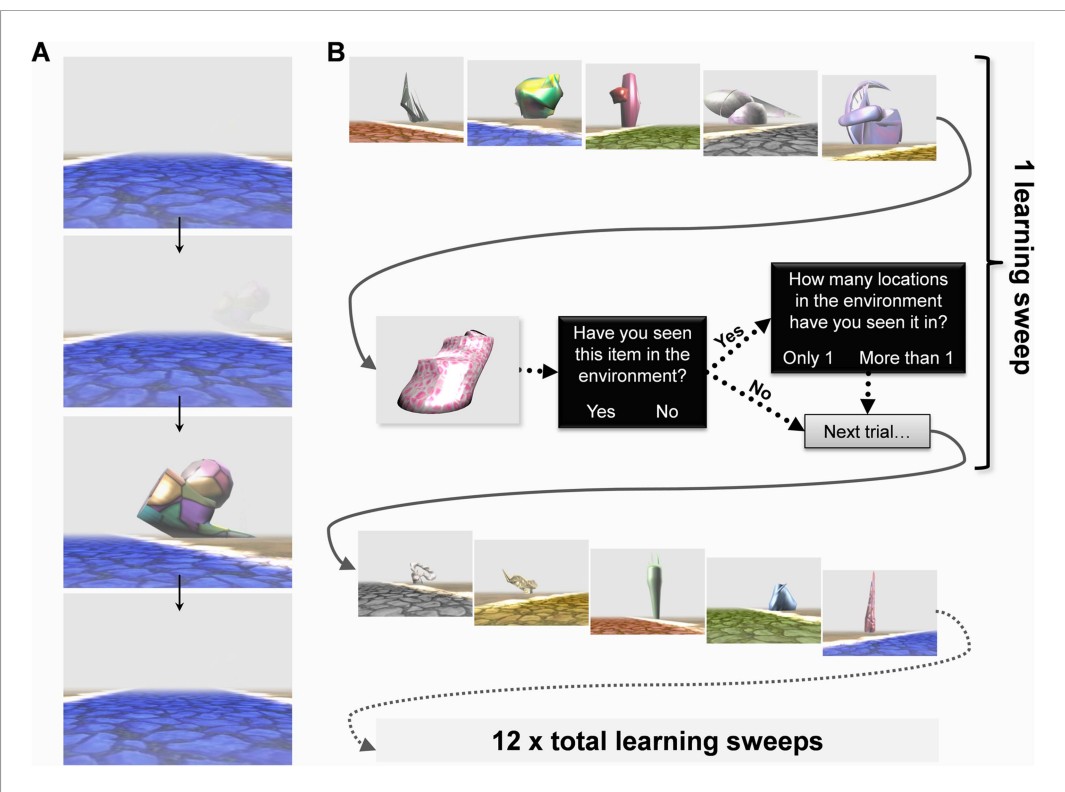

**Figure 2**. The Experimental paradigm. While undergoing functional MRI (fMRI) scanning, subjects were presented with videos travelling along the various paths. (**A**) An example sequence of video frames with a landmark emerging through the fog, the camera turning towards it before returning back to the middle of the path—see also *Video 1*. (**B**) After viewing videos of each of the five paths once, subjects answered a series of questions about individual landmarks to test their learning throughout the experiment. A learning 'sweep' consisted of one round of videos of the five paths and the questioning period which followed. There were 12 learning sweeps.

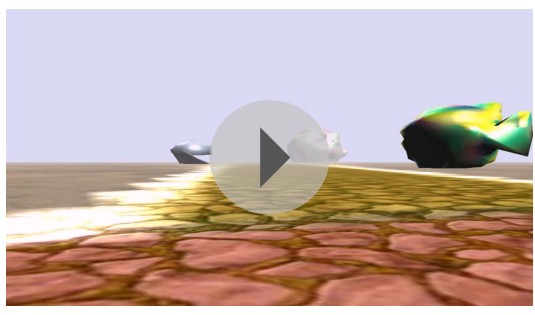

**Video 1.** This is a short clip from one of the videos which subjects viewed inside the MRI scanner when learning the environment. It demonstrates the first-person perspective presented to subjects and shows how, when a landmark emerges through the fog, the camera turns to bring it into the centre of view whilst continuing along the path. It also provides an example of what happens at an intersection. See also *Figure 2*.

We elected to show participants videos of movement through the environment during scanning rather than have them navigate volitionally because this allowed us to control exactly what they saw and ensured that all participants had the same learning experience. Participants knew that they should pay close attention to learning the environment and its layout for testing post-scan. Similar to previous experiments (e.g., *Janzen and van Turennout, 2004*; *Wolbers and Buchel, 2005*; *Doeller et al., 2007*; *Schinazi and Epstein, 2010*; *Auger et al., 2012*; *Konkle and Oliva, 2012*; *Auger and Maguire, 2013*; *Troiani et al., 2014*; *Chadwick et al., 2015*), we compared fMRI responses while subjects viewed images of individual, isolated landmarks displayed during the questioning periods at the end of each sweep, unless otherwise stated. Using this time period, rather than when landmarks were viewed during the navigation videos, removed potential problems associated with visual confounds (e.g., path colour) and the more unconstrained neural responses that may have been associated with the minute long learning videos.

## Results

### Behavioural analyses

During scanning, recognition memory for the landmarks was assessed in the questioning periods at the end of each learning sweep. We first wanted to establish whether or not subjects had learned to recognise the two types of landmark equally well. To do this, we performed separate linear regression analyses for permanent and transient landmarks to assess how the accuracy with which subjects recognised them changed throughout the learning phase in the scanner. We then directly compared the slopes and found that there was no difference in the rate at which subjects learned to recognise permanent and transient landmarks (mean difference in rate = 0.0084, SD = 0.063; $t_{31}$ = 0.763, p = 0.45).

We then examined recognition accuracy in each of the four learning quarters and observed a significant time (quarters) by landmark type (permanent, transient) interaction ($F_{3,29}$ = 8.045, p = 0.0005). Interrogating this result further, we found that that participants recognised permanent and transient landmarks equally in the first quarter (mean accuracy permanent landmarks = 57.7% (SD 9.7); transient landmarks = 58.1% (SD 17.7); $t_{31}$ = 0.12, p = 0.9) and final quarter (permanent landmarks = 79.6% (SD 18.3); transient landmarks = 77.3% (SD 14); $t_{31}$ = 0.55, p = 0.6). In the second quarter (permanent landmarks = 72.7% (SD 14.4); transient landmarks = 60.4% (SD 13.1); $t_{31}$ = 3.517, p = 0.001) and third quarter (permanent landmarks = 80.0% (SD 15.6); transient landmarks = 70.4% (SD 16.3); $t_{31}$ = 2.361, p = 0.02), however, participants were more accurate in their recognition of permanent than transient landmarks.

Consistent with the result in the final quarter of learning, in the post-scan testing phase, there was no difference in how well subjects recognised permanent or transient landmarks ('memorableness': permanent mean accuracy = 82.9% (SD 4.9); transient mean accuracy = 76.3% (SD 4.4); $t_{31}$ = 1.745, p = 0.09). Subjects were also accurate at identifying as novel landmarks which they had not seen before (mean = 93.0%, SD 2.3).

Post-scan, subjects also rated other features of the landmarks. These included whether they thought an item was permanent or transient, how visually salient they found them, and finally the size that they were in Fog World (see 'Materials and methods'). We compared these ratings with the corresponding actual values of permanence and size, and the salience scores from the separate initial landmark characterisation study (see 'Materials and methods') in order to test the validity of the scan subjects' ratings and to confirm whether or not subjects had successfully learned about the landmarks (full details in *Table 1*).

**Table 1.** Correlations between features of the 60 'alien' landmarks

| | Permanence: actual | Permanence: post-scan | Salience: beh'al study | Salience: post-scan | Size: actual | Size: post-scan |
|---|---|---|---|---|---|---|
| Permanence: actual | 1.000 | – | – | – | – | – |
| | – | – | – | – | – | – |
| Permanence: post-scan | **0.793†** | 1.000 | – | – | – | – |
| | <0.0001 | – | – | – | – | – |
| Salience: beh'al study | 0.087 | −0.001 | 1.000 | – | – | – |
| | 0.5 | 1.0 | – | – | – | – |
| Salience: post-scan | **0.315*** | **0.325*** | **0.314*** | 1.000 | – | – |
| | 0.01 | 0.01 | 0.02 | – | – | – |
| Size: actual | 0.000 | 0.068 | 0.088 | **0.428†** | 1.000 | – |
| | 1.000 | 0.6 | 0.5 | 0.001 | – | – |
| Size: post-scan | 0.117 | 0.093 | 0.129 | **0.749†** | **0.726†** | 1.000 |
| | 0.4 | 0.5 | 0.3 | <0.0001 | <0.0001 | – |

Beh'al = ratings that came from the initial behavioural landmark characterisation study. Correlations are shown between: mean salience scores from the initial characterisation study, the actual size and permanence of landmarks in Fog World, and ratings of permanence, salience and size from the fMRI subjects post-fMRI scan. Each cell shows the Pearson correlation r value above the corresponding p value. Significant correlations are highlighted in bold text.
*Correlation is significant at the 0.05 level (2-tailed).
†Correlation is significant at the 0.01 level (2-tailed).

Permanence ratings made post-fMRI scan were strongly correlated with the actual values ($r = 0.793$, $p < 0.0001$), indicating that subjects had successfully learned this information. Similarly, the size ratings in the post-scan session were significantly correlated with the actual landmarks sizes in Fog World ($r = 0.726$, $p < 0.0001$). Comparing the visual salience ratings from the initial landmark characterisation study and the fMRI study was particularly interesting. While the correlation between the two was significant ($p = 0.02$), the slope of the correlation was not particularly marked ($r = 0.314$). The landmarks in the characterisation experiment were viewed one at a time and in isolation (so not as part of Fog World). By contrast, there was a tendency for subjects post-fMRI scan to rate landmarks as more salient if they had been experienced in Fog World to be large ($r = 0.428$, $p = 0.001$) or permanent ($r = 0.315$, $p = 0.01$). In other words, the visual salience of landmarks (or how 'attention grabbing' they were) was not just an inherent property; it was also influenced by how and where they had been experienced within the environment.

Because we had a number of separate measures of landmark features, we then sought to establish if some of these variables loaded onto common underlying components. We therefore submitted the ratings and scores of permanence, size and salience of landmarks made by the scanning participants along with their memorableness scores and the actual permanence of landmarks to a principal components factor analysis using a varimax rotation and Kaiser normalization (see 'Materials and methods'). The features clearly separated onto four orthogonal factors which accounted for 96.4% of the variance in the data. These four factors were strongly related to the permanence, memorableness, size and salience of the landmarks (*Figure 3A*). Thus the factor analysis confirmed the presence of four independent components in the landmark features, which included permanence of landmarks as a distinct factor.

The post-scan testing session also involved subjects volitionally navigating to the locations of target landmarks within Fog World. This was a challenging test given that subjects had only been exposed to the environment (with its five paths and 60 landmarks) for the previous 40 minutes or so. Participants were first shown an image of a landmark and instructed that they would have to navigate to where they thought it was located in the environment by as direct a route as possible. On each trial, subjects were placed within a version of the environment in which there was no fog and the target landmark had been removed. They moved their way to where they thought that landmark belonged

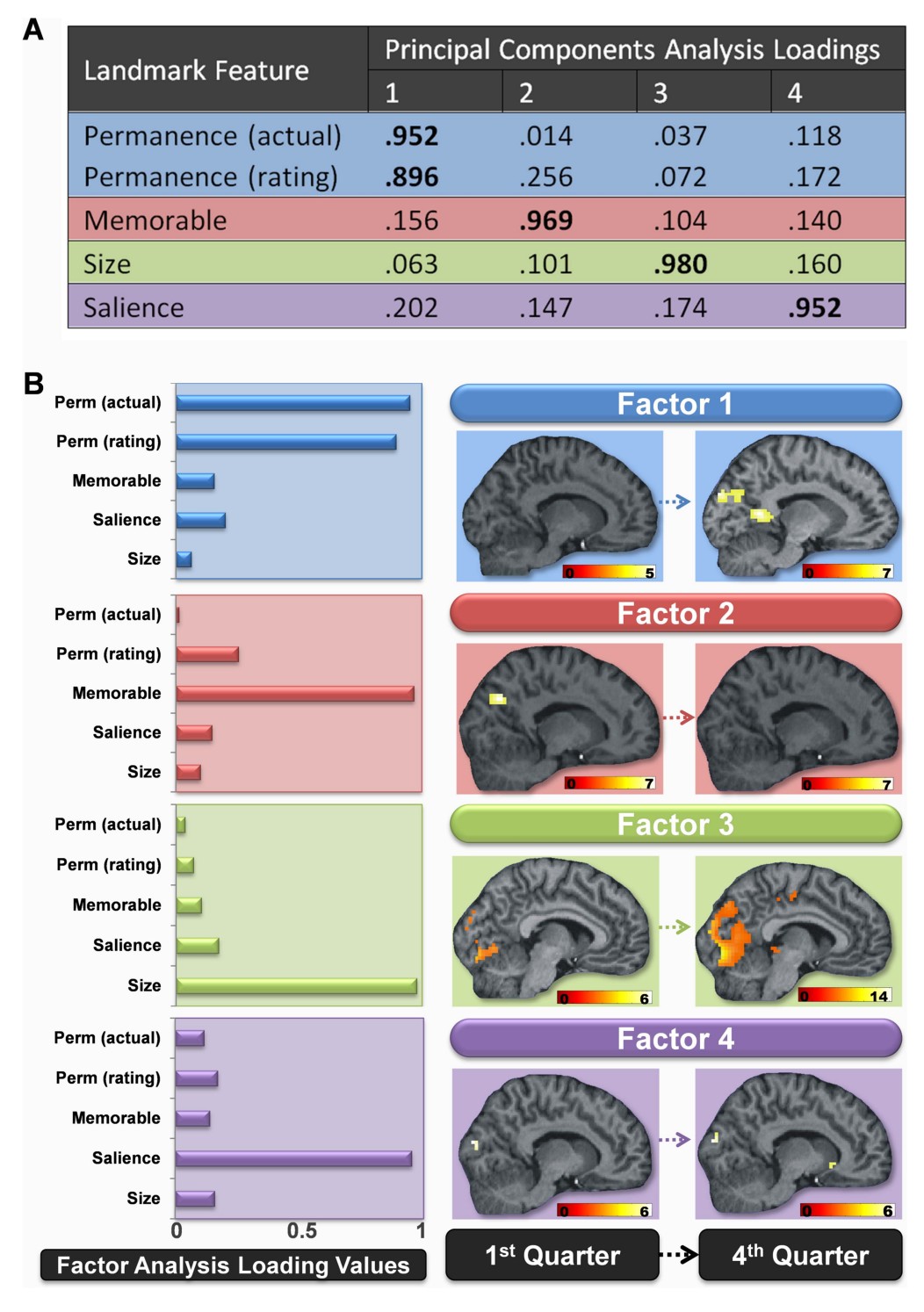

**Figure 3**. Changes in the brain regions engaged by different landmark features over the course of learning. (**A**) The loading values of each landmark feature to the four principal component factors. Values above 0.5 are highlighted in bold. Factor 1 was strongly related to landmark permanence, factor 2 to their memorableness, factor 3 to their size and factor 4 to the visual salience of landmarks. (**B**) The bar graphs to the left show how strongly each of the four factors was related to the various features rated by subjects in the post-scan debrief. The associated brain regions responding to these four factors in the first and last quarters of learning are shown to the right. All activations are shown on a structural MRI brain scan of single representative subject. Each factor's activations are shown on the

*Figure 3. continued on next page*

*Figure 3. Continued*

same sagittal slice and using a whole brain uncorrected threshold of p < 0.00001 for display purposes. The colour bars indicate the Z-score associated with each voxel.

(using the arrow keys on a keyboard) and then indicated their chosen location by pressing the space bar. There were 12 trials (nine involving permanent and three involving transient landmarks). If they thought the target landmark was transient (and so could not be placed in a single location), subjects were instructed to press the space bar and indicate that they thought it was transient. For the navigation task, each trial was scored out of 3, giving a maximum score of 36. One point was awarded for locating a landmark on the correct path, 1 point for the correct part and side of the path, and a final point was awarded if they had taken a direct route to the landmark. If they correctly identified that the target landmark was transient (and so could not be located in a single position), they were awarded 3 points.

As we expected and hoped, there was a good deal of variance in navigation task performance (mean score = 12.8, SD = 8.1). We reasoned that having such variation between individuals enabled us to examine more meaningful relationships between fMRI responses and the learning of landmark permanence. If a brain region's response was directly linked to learning of landmark permanence, one would expect activity within such a region to directly track the acquisition of this knowledge. It was therefore important to have variance in the amount of learning by subjects in order to capture such a relationship (see 'Results' section, fMRI: accounting for subject-specific learning differences). Participants performed marginally better at identifying the transient landmarks, however the scores for the transient landmarks still only constituted an average of 38.8% (SD 27.3) of the total marks (from 25% of the trials). This shows that the transient landmark trials did not have a disproportionately large effect on the overall results. The distribution of scores (out of the total marks) on the post-scan navigation task was as follows: identifying the path 33.8% (SD 16.6), identifying the part and side of path 38.1% (SD 16.5), and using a direct route 28.0% (SD 15.8). This suggests that participants were not merely making simple path colour to landmark associations, and in fact identifying the part and side of path was where more marks were scored (part vs path: $t_{31} = -2.121$, p = 0.042).

Finally, in the post-scan debriefing session we also asked participants what they had been thinking throughout the navigation videos and how they approached learning the layout of the environment. This revealed a great deal of variety in what participants were thinking. Some tended to focus on visual characteristics of the landmarks, others paid greater attention to the overall layout of the environment's paths or to individual landmarks' locations along paths. Sometimes they were thinking about the landmark in view, but on other occasions were considering what came before, or what landmark might be next. Only two out of thirty two participants made reference to specifically focussing on the permanence of the landmarks.

The variance in thought revealed by this feedback was not just large between participants but also within participants trial-to-trial making it impossible to model the fMRI time series during the videos in an accurate or meaningful way. This supported our decision to design the experiment such that the fMRI analyses focused on the inter-sweep questioning periods where landmarks were presented individually and devoid of environmental context, and may be why so many previous fMRI studies have adopted a similar approach (e.g., *Janzen and van Turennout, 2004*; *Wolbers and Buchel, 2005*; *Doeller et al., 2007*; *Schinazi and Epstein, 2010*; *Auger et al., 2012*; *Konkle and Oliva, 2012*; *Auger and Maguire, 2013*; *Troiani et al., 2014*; *Chadwick et al., 2015*).

In summary, the behavioural data showed that participants learned the basic identities of landmarks. There was a brief discrepancy in learning to recognise permanent and transient landmarks in the second and third quarters in favour of the permanent landmarks, but overall the rates of learning were similar. Participants possessed excellent knowledge of landmark permanence/ transience by the end of the scanning, as well as landmark size and visual salience. They also demonstrated some knowledge, with variance across subjects, about the overall layout of Fog World, in the volitional navigation task. We next asked what underpinned this learning in the brain.

## fMRI: permanent vs transient landmarks

We first examined the fMRI data by directly comparing permanent with transient landmarks across the whole scanning experiment. There were numerous large clusters of increased activity more for permanent than transient landmarks: one including left RSC and PHC (−21, −49, −8, z = 6.01; −12, −43, 1, z = 5.72), another including right RSC and PHC (9, −52, 4, z = 5.10; 30, −49, −5, z = 5.14), and others in left occipital cortex (−15, −88, 25, z = 5.78; −9, −85, 4, z = 4.95), left and right superior posterior parieto-occipital sulcus (POS; −3, −76, 40, z = 5.24; 15, −61, 19, z = 4.94) as well as left lateral temporal cortex (−51, −43, 4, z = 5.61), right occipital cortex (18, −88, 22, z = 5.50; 27, −82, 31) and posterior parietal cortex/precuneus (0, −34, 46, z = 4.89). No regions were more active for transient than permanent landmarks.

Having established the brain areas that were more active overall for permanent landmarks compared to transient, we then investigated when these differences arose. By the final quarter of learning (the last three learning sweeps) there were significantly greater responses to the permanent landmarks compared to the transient in right (6, −53, 5; Z = 5.41) and left (−6, −55, 10; Z = 5.90) RSC, as well as right POS (9, −73, 31; Z = 5.01) and posteriorly in the left occipital lobe (−6, −79, −8; Z = 5.00) (*Figure 4*). There were also activations in the hippocampus (−21, −28, −11; Z = 3.71) and PHC (21, −37, −14; Z = 4.12) but at a reduced threshold (p < 0.0001 uncorrected; compared with the whole-brain FWE corrected p < 0.05 reported otherwise). The increased responses to permanent landmarks were even present as early as the third quarter of scanning (sweeps 7–9 of 12) in similar regions (right RSC: 12, −51, 3; Z = 5.31; left RSC: −12, −55, 6; Z = 5.72; left POS: −6, −76, 40; Z = 4.93; left occipital: −15, −76, −11; Z = 5.53), but not in the hippocampus or PHC. There were no differences between responses to permanent and transient landmarks in either of the first two quarters of learning. No regions were more active for transient than permanent landmarks.

## fMRI: all landmark features

We next broadened the analyses to also include the other landmark features by considering how fMRI blood oxygenation level-dependent (BOLD) signals related to each of the four principal components from the factor analysis. We examined whether or not they had any identifiable neuronal correlates, and if so, how they might have evolved over the course of learning. We used the principal component scores from the factor analysis rather than the raw behavioural data because this allowed us to have fully orthogonalized regressors in the fMRI analyses. To do this, we created factor score estimates for every landmark corresponding to each of the four orthogonal principal components and then used these values to generate parametric regressors for a whole brain fMRI analysis (see 'Materials and methods'). This enabled us to examine activity that was linearly modulated by each factor.

Considering first the permanence-related factor (factor 1), namely fMRI responses associated with increasing values of this factor across the entire scanning session, the results were in line with the categorical contrast reported above: left lateral temporal cortex (−60, −40, 1; z = 6.14), left occipital cortex (−15, −91, 25, z = 5.86; −9, −64, −2, z = 4.94; −15, −73, −8, z = 4.86; −12, −31, −5, z = 4.66; −9, −85, 4, z = 4.64), left PHC (−21, −49, −5, z = 5.29), left RSC (−9, −43, 1, z = 5.03; −9, −52, 10, z = 4.61), right RSC (9, −49, 4, z = 4.90) and posterior parietal cortex/precuneus (0, −58, 43, z = 5.01). No areas showed responses associated with decreasing values of this permanence factor.

As before, having established the brain areas that were associated with increasing values of the permanence-related factor over the whole scan experiment, we then investigated when these differences arose. Increasing values of the permanence-related factor were associated with significant activations in bilateral RSC (left: −12, −55, 7; Z = 5.60; right: 9, −52, 4; Z = 5.12), as well as POS (left: −9, −91, 25; Z = 5.59; right: 9, −73, 31; Z = 5.54) in the final quarter of learning, but only right RSC in the third quarter (12, −48, 1; Z = 5.03) and no regions in either of the first 2 quarters (top of *Figure 3B*, in blue). Once again there were also increased responses in the left hippocampus (−18, −28, −11; Z = 3.99) and right PHC (24, −37, −14; Z = 4.24) but at a reduced threshold (p < 0.0001 uncorrected) in the last quarter of scanning. No areas showed responses associated with decreasing values of this permanence factor.

We then considered factor 2, the 'memorableness' factor, namely those landmarks that were better remembered, and again began by examining fMRI responses associated with increasing values of this factor across the entire scanning session. The better remembered landmarks were associated with increased activations in left occipital (−9, −79, −8, Z = 6.54), left postero-lateral

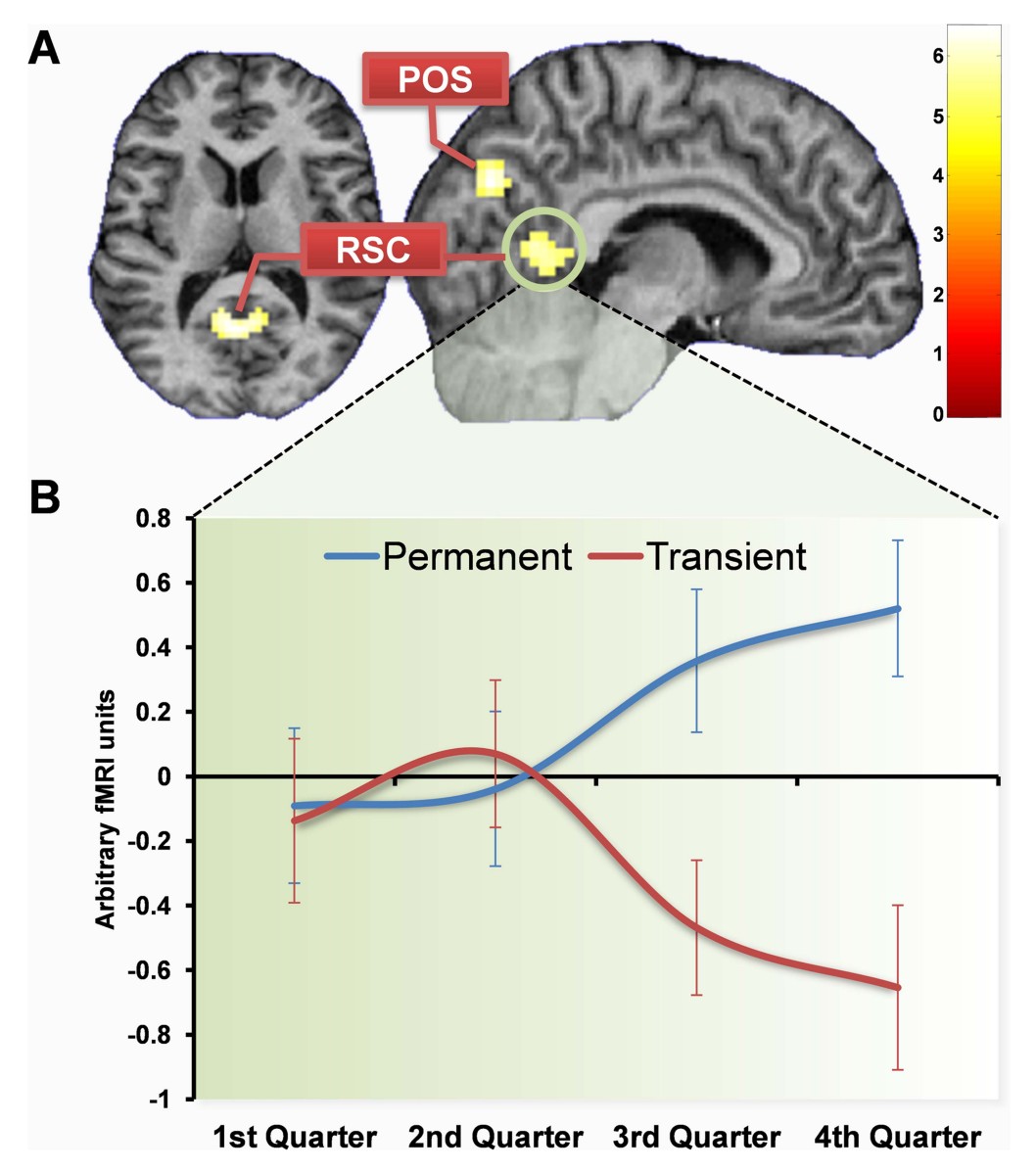

**Figure 4**. Brain regions more engaged by permanent than transient landmarks by the end of learning. (**A**) Shows activations in retrosplenial cortex (RSC) and posterior parieto-occipital sulcus (POS) at the default threshold of p < 0.05 (FWE). The colour bar indicates Z-score associated with each voxel. (**B**) Shows a plot of mean blood oxygenation level-dependent (BOLD) responses (±1 SEM) within the RSC cluster (circled in green). In the first two quarters of scanning, responses to permanent (blue) and transient (red) landmarks did not differ, but as subjects learned landmark permanence, BOLD responses increased for permanent landmarks with a corresponding decrease for transient landmarks.

parietal ($-48$, $-52$, $34$, $Z = 6.29$), and left temporal ($-63$, $-37$, $-2$, $Z = 5.81$) cortices, as well as bilateral POS (left: $-6$, $-67$, $34$, $Z = 5.87$; right: $6$, $-64$, $43$, $Z = 5.62$) and bilateral RSC (left: $-6$, $-49$, $7$, $Z = 5.10$; right: $6$, $-52$, $13$, $Z = 4.69$).

   Investigating the source of these changes showed that the more easily remembered landmarks (those with greater values for factor 2) did not produce any significant activation in the final quarter of learning. However, at the start of learning (in the first quarter), there was a greater response in POS to landmarks which participants went on to later remember better in both the right ($15$, $-70$, $31$;

Z = 5.35) and left (−9, −76, 28; Z = 5.30) hemispheres (second row of *Figure 3B*, in red). There were also similar significant activations in the middle two quarters of learning (second quarter, left: −3, −76, 40; Z = 4.94; right: 9, −67, 31; Z = 4.98; third quarter, left: −6, −64, 31; Z = 4.88; right: 3, −64, 37; Z = 4.76). Intriguingly, these bilateral regions both overlapped with those which later went on to respond to permanent items in the final quarter. No areas showed responses associated with decreasing values of the memorableness factor, either across the whole scanning session or in any of the four quarters.

Thus, the POS initially responded to memorable landmarks but then switched its response to permanent ones (*Figure 5A*). Given this overlap, we plotted the response profiles of voxels in this overlapping region for the two factors. We extracted contrast estimates of the principal eigenvariate of responses within the overlapping voxels for factor one (permanence) and two (memorableness) in each of the four scanning runs using the MarsBaR toolbox and averaged across all subjects (see 'Materials and methods'). *Figure 5B* shows the clear switch in responses within this region, with large activations initially present for the most memorable items (factor 2), but as subjects learned about the landmarks it instead (in the middle of the third quarter) became increasingly engaged by those which were permanent (factor 1).

Responses to the size and visual salience related factors (factors 3 and 4 respectively) remained constant throughout learning, with the greatest activations (associated with increased values on these factors) consistently occurring in posterior, visual areas (*Figure 3B* in green and purple respectively). For example, average responses across learning were greatest for larger landmarks in a cluster located in superior posterior parts of the occipital lobes (18, −88, 22; Z = 7.71), whereas a smaller cluster in just the right hemisphere was most active for salient landmarks (21, −91, 16; Z = 5.54). No areas showed responses associated with decreasing values of these two factors.

In summary, as subjects learned the permanence of landmarks, a representation emerged within the RSC (factor 1). POS also developed responses to permanent landmarks, but this region was also initially activated by landmarks which were subsequently better remembered (factor 2). The hippocampus and PHC were eventually more engaged by the most stable items, but later on and less strongly than RSC and POS. Perceptual features of the landmarks, their visual salience and size, were associated with tonic responses throughout learning in posterior visual areas (factors 3 and 4).

## fMRI: accounting for subject-specific learning differences

In the above fMRI analyses, we used the amount of time that people had been exposed to the environment to probe the development of neural representations of the various landmark features. However, even though subjects would inevitably have learned more about the landmarks with more exposure to them, this measure does not account for individual differences in how much participants had learned at different points throughout the experiment. This variation between individuals was important for allowing us to examine permanence-related fMRI activity in greater detail. It enabled us to identify fMRI responses which might directly track the variation in learning of landmark permanence (both between subjects and within individual people over time). This would provide more compelling evidence of whether activity in any brain region(s) might have a direct relationship with learning of landmark permanence, rather than looking at more simple, generalised associations.

To characterise the dynamics of each subject's learning more precisely, we used their scores from each sweep's questioning period to construct a range of different models of their learning-state throughout the experiment. Of the models tested (see 'Materials and methods') a Bayesian implementation of a 'state-space' model provided the best fit to the data (*Smith et al., 2007*). We used this to create subject-specific parametric regressors of each subject's estimated learning state during each sweep of the scan, examples of which are shown in *Figure 6*. We used these regressors to look for regions, anywhere in the brain, where responses matched how well a subject knew about the permanence of landmarks. The greatest activation was in the RSC (9, −58, 22; Z = 4.38), a second peak was also present in the body of the caudate nucleus (18, −10, 25; Z = 4.03). Therefore as subjects learned to distinguish permanent from transient landmarks, responses within their RSC directly reflected their knowledge of this difference.

## Connectivity analyses

We next examined changes in the functional connectivity between regions associated with learning landmark permanence using psychophysiological interactions (PPI). A PPI analysis asks whether

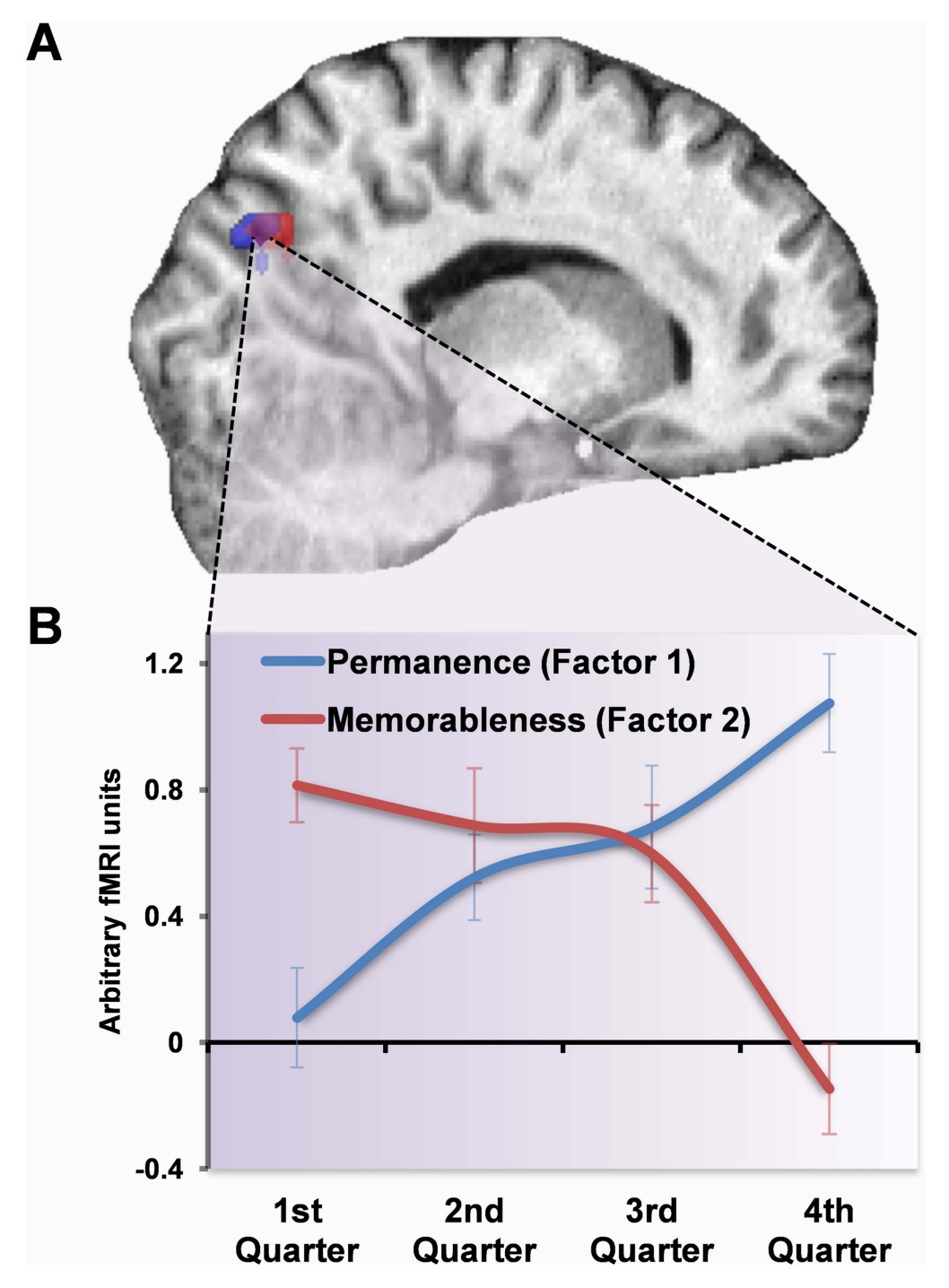

**Figure 5**. Response profile in POS. (**A**) POS responded to memorable landmarks (those with higher factor 2 values) in the first quarter of learning (red) and permanent ones (with higher values for factor 1) in the final quarter (blue). The overlap of these activations is shown in purple. (**B**) The response profile of the overlapping (purple) voxels for the two factors throughout whole scan. Responses were initially greater for memorable landmarks but then switched over the course of learning to eventually become responsive to permanence. Plots show mean BOLD responses ±1 SEM. Activations are shown on a structural MRI brain scan of single representative subject at the default threshold of $p < 0.05$ (FWE).

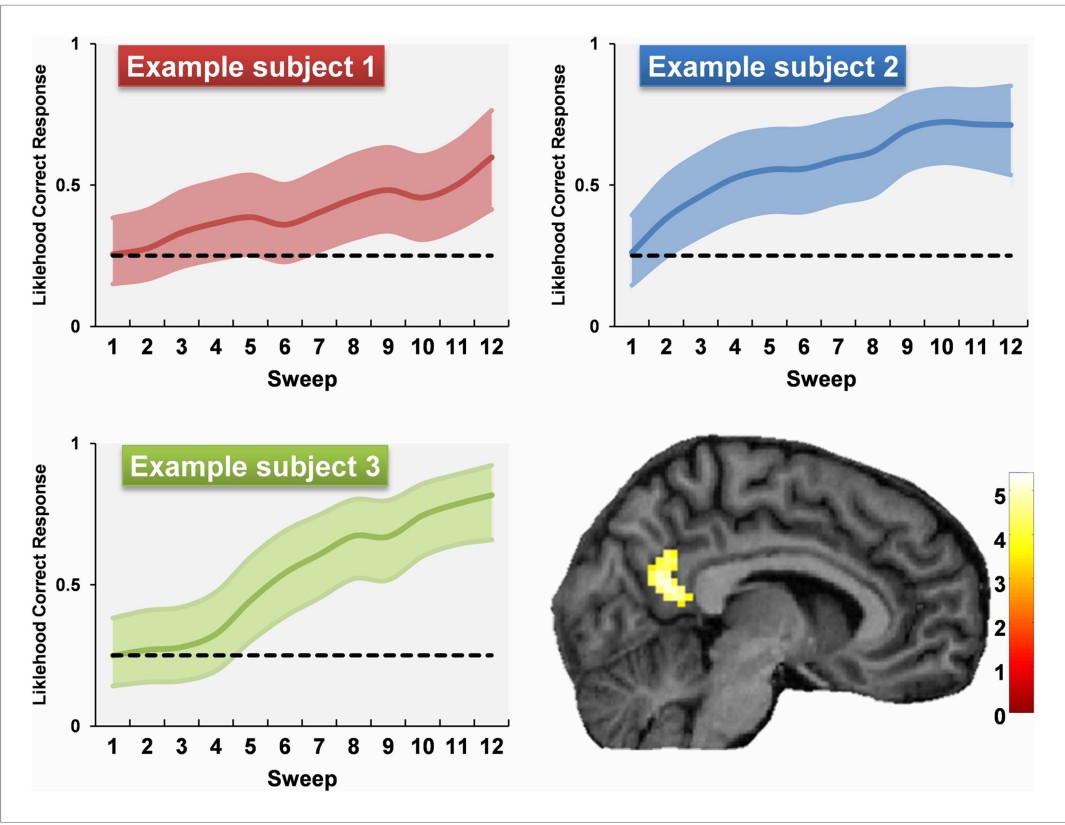

**Figure 6**. Examples of permanence learning curves and the associated fMRI response. Data from three examples subjects are shown. Learning curves were calculated and used to create subject-specific parametric regressors corresponding to the amount of permanence knowledge acquired throughout the scan. A whole brain comparison of fMRI responses to permanent vs transient landmarks according to how well subjects knew their permanence revealed responses only in RSC which were directly related to these curves. The learning curves show the estimated learning state (coloured line) and the 95% confidence interval (coloured shaded area). The activation is shown on a structural MRI brain scan of single representative subject using a whole brain uncorrected threshold of p < 0.001 for display purposes. The colour bars indicate the Z-score associated with each voxel.

anywhere in the brain has a stronger relationship with a seed region during one condition compared to another. In this instance, we performed separate whole brain PPI analyses using the parts of RSC and POS which responded to landmark permanence (*Figure 4*) as seed regions. In the second half of learning both regions showed increased functional coupling with the hippocampus when viewing permanent compared to transient landmarks. For both RSC (right RSC: 21, −16, −23; Z = 4.23; left RSC: −30, −10, −26; Z = 3.42) and POS (right POS: 30, −7, 20; Z = 3.70; left POS: −30, −13, −20; Z = 3.89), this greater functional connectivity was with anterior parts of the hippocampus bilaterally. Neither RSC nor POS showed any differences in connectivity related to permanence during the first half of learning; these only emerged as subjects learned the stability of landmarks. Moreover, there were no other brain areas that showed functional connectivity with RSC and POS.

To directly assess changes in connectivity associated with learning of permanence, we performed a further PPI analysis using the subject-specific permanence learning models. Using the same parts of RSC where greater responses emerged as subjects learned the permanence of landmarks as a seed region (*Figure 6*), the greatest increase in functional coupling developed with the left hippocampus (−30, −28, −14; Z = 3.99). Thus, as subjects learned the permanence of landmarks, their RSC not only developed greater responses to the permanent landmarks but also increased its functional connectivity with the hippocampus. In other words, the more that subjects learned about landmark permanence, the more their RSC-hippocampal functional coupling increased when viewing permanent landmarks.

## Representations related to knowledge of permanent landmark locations

We next sought to go beyond features of the landmarks and consider whether information about landmark *locations* was present within different brain regions. This required us to assess the multi-voxel representations in relation to a continuous variable (i.e., how much individuals knew about permanent landmark locations). Widely-used approaches based on linear support vector machines, such as multi-voxel pattern analysis, can only be used to make categorical classifications, and so were not appropriate for our purpose. We therefore employed an alternative type of multivariate analysis method known as multivariate Bayes (MVB). This is a model-based decoding method (*Friston et al., 2008*; *FitzGerald et al., 2012*; *Chadwick et al., 2014*) which compares competing hypotheses about the mapping between multi-voxel response patterns to a psychological target variable using a hierarchical approach known as parametric empirical Bayes (see 'Materials and method'). Specifically, we used MVB to look for patterns of voxel activity within permanence responsive regions which mapped onto knowledge of permanent landmark locations as assessed in the post-scan navigation test.

We reasoned that a representation relating to knowledge of permanent landmark locations would be strongest while subjects explicitly viewed them in that location. Therefore, unlike our other fMRI analyses, we examined fMRI responses during the learning videos, rather than during the questioning period images in which landmarks were isolated and devoid of more complex spatial information. We looked for patterns of multi-voxel activity, while people viewed permanent landmarks in situ, which related to how well they were able to subsequently locate them in the post-scan navigation task.

RSC, POS, hippocampus and PHC had all been engaged by permanent items in some shape or form during the experiment so we focused these regions. RSC, hippocampus and PHC were defined independently using bilateral anatomical masks delineated by an experienced researcher, not involved in this project, guided by *Duvernoy (1999)* and *Vann et al. (2009)* on an averaged structural brain scan from a different set of n = 30 participants. For the one permanence-responsive region which did not relate to an easily defined anatomical locus—namely the POS—we used the cluster of voxels which were activated there in the main contrast of permanent vs transient landmarks outlined at the start of the fMRI results above.

RSC, POS and PHC did not have any activity related to knowledge of permanent landmark locations at any point throughout the scanning experiment (*Figure 7*). Similarly, there were no significant results for the hippocampus during the first three quarters of learning. However, in the final quarter of the learning period, hippocampal responses emerged which were significantly related to the amount of information about the locations of permanent landmarks (log model evidence = 12.8; posterior probability = 1.0). Using the permutation function within MVB, with 100 samples, the hippocampal result in the final quarter of scanning gave a significant randomisation p value (p = 0.0396,), whereas all others were not significant.

In summary, in contrast to the RSC which responded to accruing knowledge of landmark permanence per se, by the end of the learning phase, activity patterns within the hippocampus mapped onto how much subjects knew about where those permanent landmarks were located within Fog World.

## Discussion

In this study we set out to ascertain how knowledge of landmark features, including permanence, but also properties such as size and visual salience, evolved during in situ environmental learning in adult humans, where no prior knowledge existed about the landmarks. We sought to identify the brain areas that supported this de novo learning, and to pinpoint when they came online, and what aspect of the environment each responded to. Finally, we considered how this information was used in building an overall environmental representation. By designing a bespoke VR environment and using simulated fog to completely control exposure therein, we were able to track learning within this context during fMRI in a way that has not been reported before.

We found that as the 'alien' landmarks were learned, RSC became selectively engaged by non-moving, permanent landmarks and not those which constantly changed their location. Furthermore, modelling how much individual subjects knew about the permanence of landmarks throughout the course of learning revealed that this was directly related to activity in RSC. The POS initially responded to the most memorable landmarks, but as more was learned about them, it switched to instead

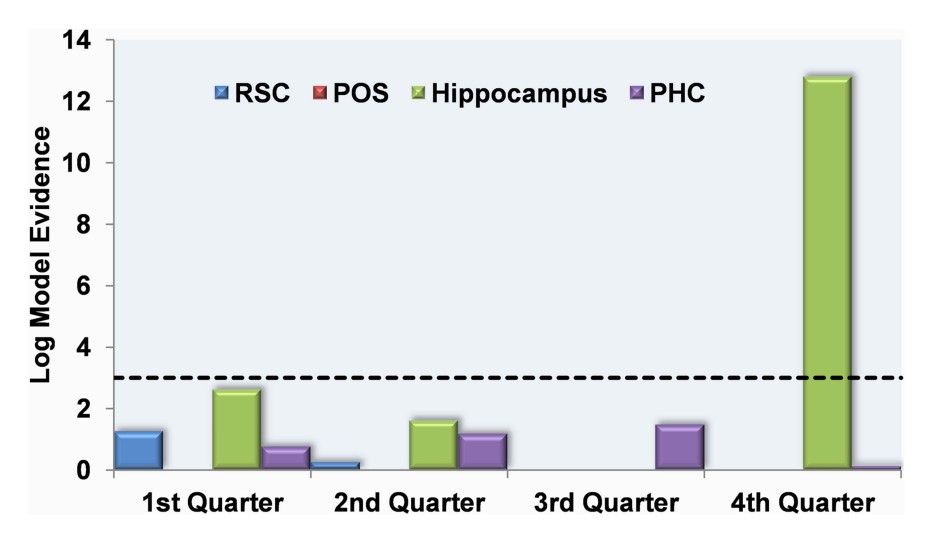

**Figure 7**. Multivariate Bayes analysis of responses which mapped onto knowledge of permanent landmark locations. The log model evidence values for response patterns within the RSC (blue), posterior POS (red), hippocampus (HC; green) and parahippocampal cortex (PHC; purple) relating to knowledge of permanent landmark locations are shown in each of the four quarters of scanning. By the final quarter of learning, the pattern of activity in the HC mapped onto the amount subjects knew about where permanent landmarks were located in the environment. The dashed black line indicates the threshold at which log model evidence values are considered to be strong (see 'Materials and methods').

become engaged by the permanent items. For perceptual features, like size and visual salience, areas within occipital cortex were engaged, and activity remained constant throughout scanning. The hippocampus was eventually activated by the permanent landmarks at the end of the scanning experiment. At the same time as this hippocampal response to permanent landmarks emerged, anterior hippocampus also showed increased functional coupling with the RSC, and activity patterns within the hippocampus mapped onto how much subjects knew about where the permanent landmarks were located within Fog World.

## Representation of landmark permanence in RSC

Given the short timescale over which subjects learned the permanence of completely novel landmarks within an alien VR world, it is notable how strong the RSC permanence representations were. It did not even take the whole scanning session for RSC to develop robust and selective sensitivity to the stable landmarks. This demonstrates the remarkable adaptability of the process and hints at it being fundamental for learning about and orientating within our surroundings. Reassuringly, there was no difference in recognition of the permanent and transient landmarks in the first or final quarters. There was a brief discrepancy in the recognition of permanent and transient landmarks in the second and third quarters of learning, in favour of the permanent landmarks. We are not sure why this is the case, but speculate that perhaps when faced with entirely novel items, we remember them better if they are in a consistent setting.

In this experiment participants were aware from the initial task instructions that some landmarks would move and other would not, and they also realised that this knowledge was probed during the inter-sweep questioning. Consequently, it could be argued that this awareness of landmark permanence may have influenced responses in RSC. In the post-scan debriefing session, however, only 2 out of 32 participants made reference to specifically focusing on the permanence of landmarks during the navigation videos. This shows that participants were not overwhelmingly concerned with the permanence of landmarks, but rather seemed to instead be focused on their overall goal, namely the requirement to learn the environment and its layout. Additionally, in previous work (*Auger et al., 2012*; *Auger and Maguire, 2013*), we have demonstrated that RSC responds to landmark permanence (for everyday outdoor items) when people are engaged in a completely incidental task.

Whether this is also true when landmark permanence is learned incidentally should be probed in future studies.

Previous experiments involving landmarks and VR have used every day, outdoor items as stimuli (e.g., *Wolbers et al., 2004*; *Wolbers and Buchel, 2005*; *Iaria et al., 2007*; *Baumann et al., 2010*; *Auger et al., 2012*; *Auger and Maguire, 2013*). These familiar, real-world items inevitably came with semantic and associative 'baggage' making it difficult to achieve precise experimental control over numerous different features of the items which are often correlated with one another (*Troiani et al., 2014*) and could potentially confound any conclusions drawn (*Sugiura et al., 2005*). Here, however, VR allowed us to investigate 'pure' representations of landmark permanence while this new information was freshly acquired. An additional advantage of using VR was that it provided a more ecologically relevant means with which to test the representation of landmark properties; it ensured subjects were actively engaged with the landmarks as they learned, in situ, the layout of the environment. However, one potential drawback of such a naturalistic, freely-behaving task is that it could introduce noise and variability to subjects' neural responses. This limitation was avoided by measuring fMRI activity when landmarks were viewed in complete isolation, as images presented between the learning videos. This not only ensured that subjects were focused on the specific relevant individual landmark at the necessary time, but also removed potential visual confounds which would have been present during the videos (e.g., path colour). Taken together with the prior studies, the current results highlight the flexible nature of RSC permanence representations. They can exist for items which are both real-world and alien; familiar or newly-encountered; viewed passively or when actively interacted with; and when attention is explicitly drawn, or not, to their permanence.

By scanning subjects as they learned about Fog World, we also uncovered new information about how representations of landmark properties develop before they have become fully established. At the start of the learning process, RSC was not responsive to any feature of landmarks. Indeed, it was only during the third quarter of scanning that RSC became engaged at all, specifically reacting to permanent landmarks. The same was not true, however, for the POS region. It became strongly responsive to permanent landmarks at a similar time to RSC, but unlike RSC was engaged initially by another feature of the items—their memorableness—before their permanence became apparent. This is an important distinction as it indicates that the representation that develops in RSC is not only very sensitive to permanence but it is also highly specific. In the absence of reliable information regarding the permanence of landmarks, RSC did not just respond to some other feature. This finding is notable for another reason. In the literature, RSC and POS are sometimes grouped together and called 'retrosplenial complex' (*Vass and Epstein, 2013*), with no differentiation made between the two brain areas. However, RSC and POS are cytoarchitecturally distinct and have different connectivity (*Vann et al., 2009*), and given that we have now pinpointed differences in their responsivity, we suggest caution in using the over-general term retrosplenial complex.

Modelling the learning-state of each individual subject throughout the experiment provided further insight into how RSC processes landmark permanence. Activity in the RSC was directly related to how well people knew the permanence of items. This reveals that the permanence representation that emerges is not just a simple binary response which indicates whether or not an item is known to be stable. Instead it appears to be more informative, relating to the precision with which the permanence of landmarks is known. In addition, therefore, to identifying the most stable environmental cues, the RSC could also indicate how reliable subsequent representations based upon such cues might be. Overall, these results point to sophisticated, selective and specific processing of landmarks within RSC based upon their permanence.

## Role of the POS

As noted above, superior posterior parts of POS displayed an intriguing profile of response; first activated by the items which were subsequently better remembered by subjects, and only later becoming engaged by the stable landmarks once their permanence was known. This region was not implicated in processing permanent landmarks in previous experiments (e.g., *Auger et al., 2012*; *Auger and Maguire, 2013*). So what is it about the memorable landmarks at the start and permanent landmarks by the end of learning in this study which engaged POS?

It is not entirely clear, but we can speculate about several possible reasons. For example, it could be that POS is tuned to navigationally 'useful' landmarks. Early in learning, in the absence of

knowledge about landmark permanence, the memorable landmarks may have seemed more useful. Focus would then be expected to shift to the permanent landmarks which ultimately had greater utility for navigating and orienting. Alternatively, our results could indicate POS plays a role in relating landmarks with a specific location. This view is in keeping with a previous interpretation of processing within 'retrosplenial complex' (*Vass and Epstein, 2013*), which often extends into posterior parts of the POS. At the start of the learning process, encounters with the most memorable landmarks (and the location they were experienced in) would have been particularly evident and so elicited the largest early responses. The level of activity for memorable landmarks would then diminish as those which are not fixed are repeatedly encountered in conflicting locations. The region would, at the same time, become increasingly engaged by the permanent landmarks with repeated experience of them in the same place. The large response at the start of the scan to the most memorable landmarks also explains why, unlike RSC, activity in POS was not directly related to the permanence learning-state. This interpretation also accounts for the lack of POS engagement in previous experiments (e.g., *Auger et al., 2012*; *Auger and Maguire, 2013*), where the stimuli were never associated with specific locations. Therefore, while RSC appears specialised in processing the permanence of landmarks, the response profile of POS is more consistent with it playing some role in indexing navigational utility, or perhaps relating landmarks with the discrete locations where they have been encountered.

## Role of the hippocampus

Another region which responded to permanent landmarks in the present study was the hippocampus. In contrast to POS, the hippocampus was not activated by any landmarks early on, only later becoming engaged once the permanent landmarks became apparent. Furthermore, unlike RSC, activity in the hippocampus was not directly related to how well subjects had learned the permanence of landmarks. Whereas RSC tracked the acquisition of permanence knowledge, the hippocampus was only engaged by landmarks which were both known to be permanent *and* associated with a specific place.

The MVB analysis indicated that once the hippocampal representation had emerged, it contained spatial detail related to how much subjects knew about where the permanent landmarks were located. This is in contrast to the other areas where no such decoding was possible, including in POS. This does not mean that POS or other regions do not contain detailed spatial information related to navigation performance; but any such representations, if they do exist, are significantly less evident than in the hippocampus. Interestingly, while the response profile in POS may suggest it initially represents landmark-location associations, as outlined above, this was not maintained over time. Moreover, the landmark-location representations in POS may have been discrete and local to a particular landmark, while the spatial detail decoded in the hippocampus was related to the wider knowledge of the environment's overall layout and the location of all the permanent landmarks within it.

We observed increased functional coupling between RSC and hippocampus towards the end of the experiment, when the hippocampus started to come online. It seems then that the RSC may code for the stability of features within an environment, potentially providing this as an input to the hippocampus, which could in turn utilise the information to build detailed spatial representations based upon the most reliable cues. In terms of RSC-hippocampus connectivity, very few anatomical tracer studies have been conducted in humans. In the macaque monkey, RSC sends extensive cortical efferent projections to the medial temporal lobes (presubiculum, parasubiculum, entorhinal and PHC cortex; *Kobayashi and Amaral, 2007*). It receives inputs from similar regions, most notably the hippocampal formation (entorhinal cortex, subiculum, presubiculum and parasubiculum), PHC and perirhinal cortices. In particular, Area 29 receives a large majority of projections from the hippocampus (both subiculum and presubiculum) and entorhinal cortex (*Aggleton et al., 2012*). Thus, it is not surprising then that we observe functional connectivity between RSC and the hippocampus, and in particular anterior hippocampus. This is also in keeping with many studies that have reported RSC and anterior hippocampus co-activing in fMRI studies (e.g., *Zeidman et al., 2014*). In addition, inactivation of RSC in rats is associated with disruption of hippocampal place fields (*Cooper and Mizumori, 2001*), and previous studies have reported hippocampal involvement in retrieving spatial information about objects (*Save et al., 1992*; *Manns and Eichenbaum, 2009*; *Baumann et al., 2010*; *Ekstrom et al., 2011*). We did not find responses relating to the landmarks in any other brain regions, but object-centred firing has also been observed in rodent lateral entorhinal

(*Deshmukh and Knierim, 2012*) and anterior cingulate cortex (*Weible et al., 2012*), even for locations formerly (but no longer) occupied by objects (*Tsao et al., 2013*). It will be interesting for future work to explore whether similar representations may also exist in humans and how they relate to the current findings.

## Uses of the RSC permanence representation

The present study provides an alternative interpretation of previous work which found that RSC is engaged when making judgements about locations relative to stable items (*Committeri et al., 2004*; *Sulpizio et al., 2013*). These previous studies concluded that activity within the RSC reflected coding of space relative to stable landmarks. However, our findings suggest it could in fact indicate a more fundamental representation of the landmark itself, specifically that of its inherent permanence. We suggest that the elementary discrimination between stable and moving landmarks demonstrated here within RSC is used to anchor representations of surrounding space. *Sulpizio et al. (2013)* also found RSC to be sensitive to viewpoint direction within a room but not relative to unstable objects. This could be linked to the presence of head direction cells within the rodent RSC (*Chen et al., 1994*; *Cho and Sharp, 2001*), perhaps suggesting that head direction cell firing is centred upon permanent landmarks and this information is integrated within RSC.

The discrimination between permanent and transient landmarks in RSC could play a crucial role in a number of fundamental computations involving space. It may go some way towards explaining its strong, ubiquitous engagement during fMRI studies of scene processing (*Epstein, 2008*; *Howard et al., 2014*; *Zeidman et al., 2014*), while navigating (*Maguire, 2001a*; *Hartley et al., 2003*; *Spiers and Maguire, 2006*; *Spreng et al., 2009*; *Vann et al., 2009*), recalling autobiographical memories (*Maguire, 2001b*; *Gardini et al., 2006*; *Steinvorth et al., 2006*; *Svoboda et al., 2006*; *Cabeza and St Jacques, 2007*; *Spreng et al., 2009*) and imagining future and fictitious events (*Addis et al., 2007*; *Hassabis et al., 2007*; *Szpunar et al., 2007*; *Botzung et al., 2008*; *Summerfield et al., 2009*). All of these involve picturing scenes or events, with RSC possibly signalling the presence of permanent features (*Auger and Maguire, 2013*).

The specificity of the RSC permanence response may also provide insights into failures of spatial navigation. *Auger et al. (2012)* and *Auger and Maguire (2013)* found that otherwise healthy individuals who were poor navigators had difficulty reliably identifying permanent landmarks, and had reduced responses in RSC compared to good navigators when viewing permanent landmarks. In neuropsychological studies, damage involving the RSC is never focal, but nevertheless leaves patients unable to derive orientation information from landmarks which they can otherwise recognise (*Maguire, 2001a*; *Vann et al., 2009*). Metabolic and structural changes in Alzheimer's Disease/Mild Cognitive Impairment have been found to be first centred upon RSC (*Villain et al., 2008*; *Pengas et al., 2010*, *2012*; *Tu et al., 2015*), and spatial disorientation is often one of the first signs of the disease. If representations of space are founded upon permanence information from RSC, as seems to be indicated by our results, then it needs to be dependable. If this is not the case, afflicted individuals may be falling at the first hurdle, so to speak, and one can understand why spatial disorientation then arises, because upstream regions that rely on good quality input cannot perform optimally to build reliable environmental representations.

## Conclusions

We believe that the current results provide compelling evidence that the RSC may be fundamentally concerned with coding for permanent landmarks. We further propose that this may account for its involvement in scene processing, navigation, autobiographical memory and future-thinking, providing input for upstream areas such as the hippocampus to then construct models of the world based upon reliable cues (*Maguire and Mullally, 2013*; *Zeidman et al., 2014*). Future studies will be needed to establish the boundaries within which the RSC operates, for instance, does its permanence coding only function within the spatial domain? The mechanisms underpinning the permanence response we report here also needs to be determined. While head direction cells have been found in rodent RSC (*Chen et al., 1994*; *Cho and Sharp, 2001*), it has been estimated that these only account for approximately 8% of cells (*Chen et al., 1994*). What are the other 92% of cells in RSC doing, and how does this relate to landmark permanence? Intracranial recordings from human RSC have also been recently reported, showing increased responsivity to autobiographical memory retrieval (*Foster et al.,*

*2013*). We hope that the current results will encourage those conducting electrophysiological recording in humans and non-humans to explore the RSC and in particular its response to landmark permanence. By giving RSC its chance to be centre stage, we believe that new and important insights into how the brain performs the fundamental computations involved in representing and adapting to changes in the world will be forthcoming.

## Materials and methods

### Landmark characterisation experiment

Ten subjects (five female, mean age 28 years, SD 4.8) took part in a landmark characterisation study. None of these subjects took part in the subsequent fMRI study. All were healthy, right-handed, highly proficient in English, and had normal vision. Each participant gave written informed consent for participation in the study, for data analysis and for publication of the study results. 'Materials and methods' were approved by the University College London Research Ethics Committee.

We created 134 unique 3D 'alien' items to be used as potential landmarks in Fog World (examples in *Figure 1A*). The landmarks were made with the animation software Blender 2.61 (Blender Foundation, Amsterdam, Netherlands, http://www.blender.org/). We then had the subjects characterise the following features of this novel set of landmarks:

- Salience ('*To what extent does this item grab your attention?*'; 5 point scale: 1 = Not at all… 5 = Very much).
- Other associations ('*Does this remind you of anything?*'; Yes/No).
- Likeableness ('*How do you feel about this item?*'; Like/Dislike).
- Animateness ('*Does this item look like it could be alive or not?*'; Alive/Not Alive).
- Memorableness ('*Have you already seen this item?*'; Yes/No—answered having seen the items numerous times while rating the other features).

Using these ratings, we selected two groups of 30 landmarks each (from the original set of 134) to be used as the permanent and transient landmarks within the VR environment. These object groups were carefully selected to ensure that they did not differ in terms of any of the features rated (t-tests of the two landmark groups: Salience: $t_{58} = 0.669$, $p = 0.51$; Other associations: $t_{58} = 0.000$, $p = 1.0$; Likeableness: $t_{58} = 0.312$, $p = 0.76$; Animateness: $t_{58} = -1.089$, $p = 0.28$; Memorableness: $t_{58} = 0.247$, $p = 0.81$). The landmarks groups did not differ on two other factors—the actual size of the items (when in the VR environment: Small/Medium/Large; [$t_{58} = 0.000$, $p = 1.0$] and mean spatial frequency [$t_{58} = -0.562$, $p = 0.58$]). Having selected the two groups of landmarks, one was randomly allocated as the permanent set of landmarks and the other as the transient set.

### fMRI experiment participants

32 different subjects (16 female, mean age 23.7 years, SD 2.4) took part in the fMRI study. All were healthy, right-handed, highly proficient in English, and had normal vision. Each participant gave written informed consent for participation in the study, for data analysis and for publication of the study results. 'Materials and methods' were approved by the University College London Research Ethics Committee.

### Fog World

Fog World was created using the jMonkeyEngine 3.0 beta game engine (http://jmonkeyengine.org), Java JDK 1.6 (Sun Microsystems, Santa Clara, California) and Blender (Stichting Blender Foundation, Amsterdam). The world (*Figure 1*) contained five different coloured intersecting straight paths (yellow, red, grey, blue and green). Each path had 12 landmarks (six permanent, six transient) evenly distributed alongside it. A trial consisted of travelling along one of these paths. There were 60 trials in total, with the five paths being travelled 12 times each (i.e., once per learning sweep). Permanent landmarks remained in the same location on each trial, whereas transient landmarks appeared in a different location on every exposure. The locations in which all 60 landmarks appeared on each of the 60 trials were meticulously designed so that permanent and transient landmarks were equally distributed either side and along the whole length of each path. This ensured that the permanent and transient landmarks, as well as being matched for their perceptual features (size, visual salience, and other features—see above), were placed in equivalent locations within the environment.

Having determined the identities and precise locations of permanent and transient landmarks within Fog World on all 60 trials, we created a video for each trial to be presented to subjects while they underwent fMRI scanning. Each video took a first person perspective travelling along one of the paths. In these videos, the environment was covered in a shroud of fog to restrict the field of view and ensure close control over the exposure subjects had to all the landmarks. On each trial, the camera travelled along a path in a straight line. When a landmark emerged out of the fog, the camera turned to bring the landmark into the centre of the screen, where it was positioned for 2 s, the camera then panned back to the middle of the path as it continued travelling forwards (*Figure 2A* from top to bottom; see also *Video 1*). The paths were always travelled in the same direction, with the same start and end point each time. During scanning, the ordering of trials along the five different paths within each learning sweep was pseudorandomised so there were no biases in when the paths were travelled relative to each other.

To encourage subjects to learn an integrated representation of the whole environment, the paths intersected one another. Each path intersected with two others (*Figure 1C*). The first intersection was located three landmarks after the start of the path and the second was three landmarks before the end, with six landmarks between the two intersections. When the videos came to one of these intersections, the camera turned either left or right and the fog cleared enough to reveal three landmarks on the adjoining path. After 3 s, the landmarks were obscured by the fog again with the camera returning to the centre while continuing along the route. There were equal numbers of left and right turns at each intersection throughout the whole experiment and the ordering of the turns was pseudorandomised to ensure it was not predictable. The number of times each landmark was viewed during one of these intersection turns was also controlled so that overall exposure to all the landmarks remained identical. These 60 videos (corresponding to the 60 trials) were each approximately 1 min in length.

## Task and procedure

Before scanning, subjects were shown an example trial (containing landmarks and a path which did not appear during the main experiment) to familiarise them with the general format of the main fMRI task. The main fMRI task has been described earlier. To summarise, once all five paths had been travelled once, there came a questioning period to gauge how much information subjects had learned by that point in the experiment. The combination of a 13 landmark questioning period and videos of the five different paths preceding it are referred to as a learning 'sweep'. In the questioning period between learning sweeps, the ordering of the three types of landmark (permanent, transient or unseen) was also pseudorandomised. There were a total of 12 learning sweeps throughout the experiment (divided into four scanning runs, or 'quarters', comprising three sweeps each). Each 3-sweep scanning run lasted 15–20 min and subjects could take a short break (while remaining in the scanner) between scanning runs when necessary.

Once out of the scanner after the learning experiment had concluded, subjects were shown images of individual landmarks (all 60 from the environment and 26 previously unseen landmarks) and indicated whether or not they recognised them from the environment ('*Do you remember seeing this item in the environment?*', Yes/No—the memorableness measure). After that, questions were only asked about the landmarks from the environment. Participants first rated the permanence of the environment's landmarks ('*How many positions in the environment do you think this item was in?*', Only 1/Many); next they rated the salience of each landmark ('*To what extent does this item grab your attention?*', Not at all/A bit/A lot) and finally the size that landmarks were in the environment ('*What size is this item?*', Small/Medium/Large). A different randomised order of landmarks was used for each of these questions.

A final active navigation task provided a thorough examination of how well participants had learned the layout of the whole environment. This task and scoring are described in the main text.

## Behavioural analyses

During scanning, two ratings were collected from subjects to gauge how well they recognised landmarks and knew their permanence. We assessed the rates at which subjects came to recognise permanent and transient landmarks. To do this, we performed separate linear regression analyses for permanent and transient landmarks to assess how the accuracy with which subjects recognised them

changed throughout the learning phase in the scanner. Using a paired t-test (threshold $p < 0.05$) we then directly compared the slopes using these linear estimates across all subjects in order to establish whether or not subjects had learned to recognise the two types of landmark equally.

The principal components factor analysis was conducted using the mean ratings for each feature of all 60 landmarks from every subject; it used a varimax rotation and Kaiser normalisation. We then generated orthogonal factor score estimates using the Anderson-Rubin method for use in a whole brain fMRI analysis. The factor analysis and all statistical tests were performed using SPSS version 20 (http://www.spss.com).

## Scanning parameters and preprocessing

T2*-weighted single-shot echo-planar images with BOLD contrast were acquired on a 3T Magnetom Allegra head-only MRI scanner (Siemens Healthcare, Erlangen, Germany) operated with the standard transmit-receive head coil. fMRI data were acquired across four sessions with a sequence which was optimized to minimize signal dropout in the medial temporal lobe and used a descending slice acquisition order with a slice thickness of 2 mm, an interslice gap of 1 mm, and an in-plane resolution of 3 × 3 mm (*Weiskopf et al., 2006*). 48 slices angled at −45˚ to the anterior–posterior axis were collected covering the entire brain, with a repetition time of 2.88 s, 30 ms echo time and 90˚ flip angle. A 3D MDEFT T1-weighted structural scan was also acquired for each participant with 1 mm isotropic resolution (*Deichmann et al., 2004*). The first 6 'dummy' volumes from each of the four sessions were discarded to allow for T1 equilibration effects. FMRI data were analysed using SPM8 (www.fil.ion.ucl. ac.uk/spm). Images were realigned and unwarped using field maps which were acquired with a double-echo gradient field map sequence (TE = 10 and 12.46 ms, TR = 1020 ms, matrix size 64 × 64, with 64 slices, voxel size = 3 mm$^3$) and then normalised to a standard EPI template in MNI space with a resampled voxel size of 3 × 3 × 3 mm and smoothed using an 8 mm FWHM Gaussian kernel.

## fMRI analyses: permanent vs transient landmarks

We compared fMRI responses while subjects viewed images of individual, isolated landmarks displayed during the questioning periods at the end of each sweep. Our primary interest was in seeing whether a neural representation of landmark stability emerged for previously unseen items over the course of the scanning experiment. We therefore directly contrasted fMRI BOLD responses to permanent and transient landmarks in the whole brain. We first considered responses across the whole scanning session, and in a second analysis divided the scanning experiment into quarters (which corresponded to the four scanning runs each consisting of three learning sweeps) in order to assess changes as subjects learned about the items. Potential issues associated with incidental changes in the BOLD signal over time were avoided by specifically comparing changes in the difference between permanent and transient landmarks. Permanent and transient landmark regressors were convolved with the haemodynamic response function. A separate regressor was created for the learning video time periods. This, along with participant-specific movement regressors were treated as covariates of no interest. We calculated subject-specific parameter estimates pertaining to each regressor of interest (β) for each voxel. Second level random effects analyses were then run using one-sample t-tests on these parameter estimates. We report all fMRI results at a whole brain threshold of $p < 0.05$ (FWE), unless otherwise stated.

## fMRI: all landmarks features

We also examined fMRI responses in relation to the four factors from the principal components analysis. As above, we first analysed fMRI BOLD responses across the whole scanning session, and in a second analysis divided the scanning experiment into quarters. We created parametric regressors from the orthogonal factor score coefficients for every landmark for each of the four principal components using the Anderson-Rubin method. Parametric regressors from these scores were then entered into a whole brain GLM fMRI analysis. This enabled us to examine activity that was linearly modulated by each factor. These parametric regressors were each convolved with the haemodynamic response function. Analyses were then run in the same way as for the permanent vs transient landmark comparison described above.

We used the MarsBaR toolbox for analysing responses within the overlapping voxels for the permanence and memorableness factors. MarsBaR is an SPM toolbox which extracts fMRI data from

specified regions of interest. We defined the region of interest (purple in *Figure 5A*) as the contiguous voxels which showed significant activation for both the memorableness factor in the first quarter of learning and the permanence factor in the fourth quarter. For each subject, we then extracted the principal eigenvariate of responses within this region for the memorableness and permanence factor parametric regressors, in each of the four quarters of the scanning period. The mean $\pm 1$ SEM of these subjects' responses are plotted in *Figure 5B*.

### fMRI: accounting for subject-specific learning differences

The 'state-space' learning models were created with the MATLAB- and WinBUGS-based software provided at http://www.neurostat.mit.edu. We compared the mean squared error (MSE, in arbitrary units) and percentage variance explained by three different types of learning model: a 'state-space' model estimated by maximum likelihood using an expectation maximisation algorithm (*Smith et al., 2004*), a 'state-space' model estimated by a Bayesian approach (*Smith et al., 2007*) and a moving average of accuracy across each sweep and the sweeps immediately preceding and following it. A Bayesian implementation of a 'state-space' model provided the best fit (*Smith et al., 2007*) by both accuracy measures (MSE = 204, SD = 88; $r^2 = 64$) than the 'state-space' model estimated by maximum likelihood (MSE = 226, SD = 85; $r^2 = 44$) and the moving averages model (MSE = 221, SD = 86; $r^2 = 36$).

As the state-space model estimated by a Bayesian approach (*Smith et al., 2007*) provided the best fit of the data, we used this to create subject-specific parametric regressors of each subject's estimated learning state during each sweep of the scan for use in a whole brain GLM analysis. Separate regressors were created for permanent and transient landmarks so that we could contrast responses to the two types of landmark in direct relation to how well people knew the permanence of landmarks.

As SPM automatically mean centres parametric regressors within each scanning block, we concatenated the four sessions into one and added extra regressors to model the mean signal for each session. Parametric regressors are additionally mean centred by SPM at the first-level, so in order to accurately reflect between-subject differences in the overall extent of learning across the whole experiment, we added their overall performance on the post-scan navigation test as a second-level covariate of interest. Significant clusters are reported at a whole brain uncorrected threshold of p < 0.001 for the RSC and p < 0.05 FWE corrected for the rest of the brain. We chose this statistical threshold for the RSC given the more subtle nature of this specific contrast (compared to the simpler categorical comparison of all permanent with all transient landmarks) and our specific prior hypotheses about RSC processing landmark permanence. However, it should be noted that the RSC activation was also significant after applying small volume correction within a bilateral RSC mask (threshold FWE corrected p = 0.01).

### Connectivity analyses

Functional connectivity was examined using the gPPI toolbox (*McLaren et al., 2012*). We looked for changes in the functional connectivity of regions associated with learning landmark permanence. The seed regions and contrasts used for these analyses were all based upon the corresponding univariate whole-brain comparisons described above. First, for any regions responsive to landmark permanence, we looked for brain areas with which they showed increased functional coupling for permanent compared to transient landmarks. Early and late parts of the scanning session were compared separately (learning sweeps 1–6 and 7–12 respectively). We analysed the two halves instead of four quarters in order to increase the number of trials and power with which to detect these potentially subtle effects. In a second connectivity analysis, implemented in the same way, we used a seed region and contrast which additionally accounted for inter-individual differences in learning.

### Representations related to knowledge of permanent landmark locations (MVB analysis)

The MVB analysis uses the same design matrix as a standard univariate SPM analysis, with columns for experimental variables of interest as well as regressors of no interest. A contrast is then specified (in this instance from subjects' scores on the post-scan navigation task) and a 'target' variable is derived from this after accounting for potential confounds (e.g., head movement). The patterns of voxel

activity within each region of interest are then fitted to this target variable, producing a model evidence value. These model evidence values are then compared to a null model to determine the log model evidence.

For each model, voxel weights are calculated which map every voxel to knowledge of permanent landmark locations. The priors and variance of the patterns of these voxel weights are used to constrain the models at a second hierarchical layer. The pattern weights are iteratively bipartitioned into subsets based upon the size of the weights; each successive partition isolates the subset with the largest pattern weights. These subsets are optimised using a greedy search with a standard variational scheme under the Laplace assumption (*Friston et al., 2007*). Once the optimal set size is reached, this final set of patterns represents the decoding model with the greatest evidence for the pattern weights. Log model evidence values therefore represent the mutual information shared by the psychological variable (in this case knowledge of permanent landmark locations) and the pattern of voxel responses within that brain region. The log evidence of different models can then be directly compared.

MVB assumes that a small proportion of voxel activity patterns make a large contribution to decoding accuracy (i.e., information is sparsely coded). We used the SPM software's default settings for the MVB analyses, with nine greedy search steps and size of successive subdivisions set at 0.5 to test for evidence of sparse representations relating to knowledge of landmark location. We modelled the whole time period that permanent landmarks were in view during the learning videos. We defined the regions of interest anatomically (with one exception—POS—see main text), as exploring responses within the whole bilateral anatomical regions rather than smaller functionally-defined clusters within them provided maximal multi-voxel information for the characterisation of representations. We report all analyses with a log model evidence value above three as significant, as is common practise (*Kass and Raftery, 1995*; *Penny et al., 2004*; *Friston et al., 2008*). We conducted further control analyses using the permutation function within MVB (with 100 samples) to check there was no bias towards the MVB procedure producing a positive result.

## Acknowledgements

We thank Imaging Support and Ric Davis for technical assistance and Karl Friston, Guillaume Flandin, Martin Chadwick and Tom FitzGerald for helpful discussions. We are also grateful to the community of the jMonkeyEngine game engine (http://jmonkeyengine.org/) for their technical advice.

## Additional information

### Funding

| Funder | Grant reference | Author |
| --- | --- | --- |
| Wellcome Trust | 101/759/Z/13/Z; 091593/Z/10/Z | Eleanor A Maguire |
| Brain Research Trust (BRT) | PhD Studentship | Peter Zeidman |
| National Institute for Health Research (NIHR) | PhD Studentship | Stephen D Auger |

The funders had no role in study design, data collection and interpretation, or the decision to submit the work for publication.

### Author contributions

SDA, Conception and design, Acquisition of data, Analysis and interpretation of data, Drafting or revising the article; PZ, Conception and design, Analysis and interpretation of data; EAM, Conception and design, Analysis and interpretation of data, Drafting or revising the article

### Ethics

Human subjects: The studies were approved by the University College London Research Ethics Committee: #1825/003 Minimum Risk Magnetic Resonance Imaging Studies of Healthy Human Cognition. Written informed consent was obtained from each participant for participation in the study, for data analysis and for publication of the study results.

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
