## [Decision Letter]

Thank you for submitting your work entitled “A central role for the retrosplenial cortex in de novo environmental learning” for peer review at *eLife*. Your submission has been favorably evaluated by K VijayRaghavan (Senior Editor), a Reviewing Editor, and two reviewers.

The reviewers have discussed their reviews with one another and the Reviewing Editor has drafted this decision to help you prepare a revised submission. While generally enthusiastic, the reviewers had several concerns that need to be addressed in a revision.

Reviewer #1:

In this manuscript, the authors employ a cleverly designed novel virtual reality (VR) paradigm to investigate how the human brain acquires knowledge about landmarks. Using unfamiliar objects without pre-experimental semantic associations, the authors provide evidence that the retrosplenial cortex (RSC) represents the permanence of landmarks. The authors report increased activation in a number of brain regions, including the RSC and the posterior parieto-occipital sulcus (POS) when showing images of permanent compared to non-permanent landmarks, which were presented at multiple locations over the course of the experiment. Further analyses revealed that this effect emerged in the second half of the experiment as participants learned about which landmarks were permanent and which were not. Interestingly, the increased RSC activation in response to permanent landmarks paralleled participants' knowledge about landmark permanence. Parametric analyses using the landmarks' factor scores obtained in a factor analysis show that RSC and POS also exhibit increased responses for better remembered landmarks. The activation profile of the POS suggests that this region responds more strongly to more memorable landmarks at the beginning of the experiment before responding more strongly to permanent compared to nonpermanent landmarks in the last quarter of the experiment. Lastly, the authors relate multi-voxel patterns in the hippocampus towards the end of the experiment to the amount of information learned about the landmarks' locations.

1) Do the authors really measure de novo spatial learning? On the one hand, the authors argue in the Introduction that de novo spatial learning has not been investigated in rodents since animals are – despite being exposed to novel environments – familiar with the general experimental procedures. But how is this conceptually different from the current study where probably most of the participants are familiar with viewing experimental stimuli (albeit unfamiliar ones) in a MRI scanner?

2) Most of the analyses are based on the brain responses during the question period following the learning videos when participants viewed landmark images. I agree with the authors that these time periods are visually better controlled compared to the navigation videos in which the landmark is visible together with the differently colored paths in the VR environment. However, the restriction of the analysis to this time period also weakens the authors' conclusions about the role of the RSC in coding for permanent landmarks during environmental learning. Since participants were explicitly asked to indicate the permanence of recognized landmarks during the question period, this might have caused them to particularly represent this feature of the landmark in question. If the role of the RSC lies indeed in the coding of landmark permanence as assumed by the authors, then a modulation of RSC activity should also be evident when landmarks are viewed during the course of the navigation videos while participants tried to learn the layout of the VR environment.

3) With the MVPA analysis, the authors try to provide evidence for hippocampal representations in the last quarter of the experiment related to knowledge about the landmarks' locations in the VR environment. What concerns me about this analysis is the low level of behavioral performance in the post-scan navigation test aiming at measuring knowledge of landmark locations. On average, participants score about 1/3 of the 36 possible points. While a trial (locating one landmark) could score a maximum of three points, the authors do not provide details about which points participants mostly obtained. It seems very likely that the majority of points were awarded for correctly identifying non-permanent landmarks. The highly-significant correlation between actual and judged landmark permanence suggests that participants were able to identify non-permanent landmarks. Additionally, a simple associative strategy might have accounted for points collected when participants replaced permanent landmarks. One of three points for a landmark was awarded for choosing the correct path. Since the five paths used in the task were colored differently, a simple association between landmark and path color might have been used to solve the task. This might explain why multi-voxel patterns in the hippocampus, whose role in associative learning is well established, mapped onto performance in the post-scan test.

Reviewer #2:

In this paper, the authors provide a novel demonstration of the role of the retrosplenial cortex in navigation and spatial memory. Participants learned a novel environment, which contained landmarks with either permanent or variable locations. The authors found that the retrosplenial cortex (RSC) was more active for permanent, as compared to transient, landmarks. Additionally, they found that the retrosplenial activity in response to the permanent landmarks increased over the course of learning, peaking in the final quarter of the experiment. Similar patterns of activity were shown in the posterior parieto-occipital sulcus (POS). These regions also demonstrated increased connectivity with the anterior hippocampus in the latter half of the study during viewing of permanent landmarks. Finally, while the RSC and POS demonstrated increased activity relating to the permanence of the landmarks, multivariate Bayes analysis revealed that hippocampal activity related to the locations of the landmarks, but only in the final stage of learning, when locations were best known.

This study is well-designed and carefully analyzed, yielding interesting novel findings about the role of the retrosplenial cortex and parieto-occipital sulcus in navigation and learning new environments. The authors use multiple analysis techniques to provide converging and generally convincing evidence for the importance of the RSC in learning landmark permanence. The manuscript is clear and well-written. I have a few suggestions for clarification and revision prior to publication.

1) For the behavioural results, the authors report that the slopes of learning were no different for permanent versus transient landmarks and neither were the final measures of recognition, but do not report whether recognition accuracy differed across the permanent and transient landmarks at time points throughout learning (i.e. when analyzed by quarter, in keeping with the imaging results). Seeing as the permanent landmarks have a numerically greater recognition rate at final test, it would be interesting to see if this difference was significant at any time throughout learning.

2) Regarding the principal components analysis, since the analysis separated the four measures into four distinct factors, it was not clear to me why the factor scores were used in later analyses rather than just the behavioural measures themselves. Could the authors explain the advantage of this? Furthermore, when describing the results of the PCA, the authors state that it “in particular highlighted permanence of landmarks as a distinct factor”. The loading scores don't seem to indicate that the permanence factor was any more distinct than the others. Are the authors referring to some other metric that was not included in the Results section, or do they just mean to highlight this, since permanence is the factor of interest in this study?

3) The authors report robust and converging evidence for the RSC and POS activity relating to the permanence of landmarks, and report that the hippocampus activity relates to knowledge of location, but only in the final quarter of learning. However, given the behavioural results (i.e. that participants learned permanence/transience very well, but navigation performance was low and variable), how much can be made of the null results regarding landmark location for the other areas and learning quarters? The authors should comment on this in the Discussion.

4) In a related vein, throughout learning, participants were asked about the permanence of the landmarks. While I acknowledge that this was an important aspect of the design in order to ascertain how well this information was learned, it raises the question of how much influence this orientation to permanence of the landmarks had on performance and neural activity. Did the authors debrief the participants in order to determine how much attention they were paying to learning the permanence of the landmarks? This should be addressed, and future studies could try to see if behavioural and neural responses are as robust if landmark permanence is only ascertained incidentally.

5) Regarding the connectivity analyses, the authors only report results for the anterior hippocampus. Did no other regions show this pattern? The Methods section indicates that “any regions responsive to landmark permanence” were tested for connectivity differences. Does this mean that only the hippocampus and PHC were tested for connectivity changes? More detail about what analyses were performed should be included. Also, is the connectivity of the anterior hippocampus to RSC consistent with its anatomical connections? Is the posterior hippocampus most closely linked anatomically to RSC?

6) The authors speculate about the role of the POS, attributing the switch from coding memorability to permanence to the subject relating landmarks to specific locations. However, since the MVB analyses did not find any evidence of POS activity relating to landmark locations, this argument seems very speculative. The authors should elaborate on their claim that, “the landmark-location representations in POS may have been discrete, while the spatial detail decoded in the hippocampus was related to the knowledge of the environment's overall layout and the location of all the permanent landmarks within it”. Is there any evidence for this? Are there other possible accounts of the patterns of POS activity? It seems that by initially tracking the most memorable landmarks, and then once permanence information is learned, switching to track this measure, the POS could be generally tuned to whatever landmarks seem the most “useful’ navigationally. Before information about permanence has been ascertained, the most memorable landmarks may be the ones that are focused on, but once the participants determine which landmarks are permanent, they may shift to focus on these in order to orient and navigate better.

---

## [Author Response]

Reviewer #1:

1) Do the authors really measure de novo spatial learning? On the one hand, the authors argue in the Introduction that de novo spatial learning has not been investigated in rodents since animals are – despite being exposed to novel environments – familiar with the general experimental procedures. But how is this conceptually different from the current study where probably most of the participants are familiar with viewing experimental stimuli (albeit unfamiliar ones) in a MRI scanner?

We agree this lacks clarity. We meant to convey the point that, besides the developmental rodent work that we highlight, there have been no systematic, dedicated neuroscientific studies in either humans or animals examining how fundamental environmental knowledge, such as that relating to landmarks and their properties, develops in the first place. We have now re-worked sections of the Introduction to make this clearer.

*2) Most of the analyses are based on the brain responses during the question period following the learning videos when participants viewed landmark images. I agree with the authors that these time periods are visually better controlled compared to the navigation videos in which the landmark is visible together with the differently colored paths in the VR environment. However, the restriction of the analysis to this time period also weakens the authors' conclusions about the role of the RSC in coding for permanent landmarks during environmental learning. Since participants were explicitly asked to indicate the permanence of recognized landmarks during the question period, this might have caused them to particularly represent this feature of the landmark in question. If the role of the RSC lies indeed in the coding of landmark permanence as assumed by the authors, then a modulation of RSC activity should also be evident when landmarks are viewed during the course of the navigation videos while participants tried to learn the layout of the VR environment*.

This is an important point and one to which we gave considerable thought when designing the experiment. Ideally, we would have focused the analysis on the navigation movies, but it was not possible to know precisely what subjects were thinking even at specific points such as when landmarks were in view. For instance, in the post-scan debriefing session we asked participants what they had been thinking throughout the navigation movies and how they approached learning the layout of the environment. This revealed a great deal of variety in what participants were thinking. Some tended to focus on visual characteristics of the landmarks, others paid greater attention to the overall layout of the environment’s paths or to individual landmarks’ locations along paths. Sometimes they were thinking about the landmark in view, but on other occasions were considering what came before, or what landmark might be next. Only two out of thirty two participants made reference to specifically focusing on the permanence of the landmarks.

Firstly, this feedback shows that participants were not overwhelmingly focused on the permanence of landmarks, even though they were aware from the initial task instructions that some landmarks would move and others would not, and realised this knowledge was probed during the inter-sweep questioning. It seems participants kept their overall goal in mind, namely the requirement to learn the environment and its overall layout.

Secondly, the variance in thought revealed by the debrief feedback was not just large between participants but also within participants trial-to-trial, making it impossible to model the fMRI time series during the movies in an accurate or meaningful way. It is for this reason we considered it important to use images which were devoid of environmental context as the focus of our analyses, and may be why so many previous fMRI studies have adopted a similar approach (e.g. [38]; [84]; [19]; Schinazi et al., 2010; [41]; [73]; [10]). We are therefore confident that our analysis was the most appropriate to employ in this instance and that what we captured in our results was the meaningful evolution of landmark and environmental knowledge. In the revised manuscript, we include a more detailed rationale for our approach, provide details of participants’ feedback from the debriefing session, and cite all of the papers above so the reader can appreciate that this approach is widespread in the field.

*3) With the MVPA analysis, the authors try to provide evidence for hippocampal representations in the last quarter of the experiment related to knowledge about the landmarks' locations in the VR environment. What concerns me about this analysis is the low level of behavioral performance in the post-scan navigation test aiming at measuring knowledge of landmark locations. On average, participants score about 1/3 of the 36 possible points. While a trial (locating one landmark) could score a maximum of three points, the authors do not provide details about which points participants mostly obtained. It seems very likely that the majority of points were awarded for correctly identifying non-permanent landmarks. The highly-significant correlation between actual and judged landmark permanence suggests that participants were able to identify non-permanent landmarks. Additionally, a simple associative strategy might have accounted for points collected when participants replaced permanent landmarks. One of three points for a landmark was awarded for choosing the correct path. Since the five paths used in the task were colored differently, a simple association between landmark and path color might have been used to solve the task. This might explain why multi-voxel patterns in the hippocampus, whose role in associative learning is well established, mapped onto performance in the post-scan test*.

Thank you for this interesting suggestion. We have now conducted further analyses to address these two points. Firstly, participants performed marginally better at identifying the transient landmarks, however the scores for the transient landmarks still only constituted an average of 38.8% (SD 27.3) of the total marks (from 25% of the trials). This shows that the transient landmark trials did not have a disproportionately large effect on the overall results.

Secondly, the distribution of scores (out of the total marks) on the post-scan navigation task was as follows: identifying the path 33.8% (SD 16.6), identifying the part and side of path 38.1% (SD 16.5), and using a direct route 28.0% (SD 15.8). This suggests that participants were not merely making simple path colour to landmark associations, and in fact identifying the part and side of path was where more marks were scored (part vs path: t_31_ = -2.121, p = 0.042). We have now incorporated these extra details into the revised manuscript (see the Results section).

Reviewer #2:

*1) For the behavioural results, the authors report that the slopes of learning were no different for permanent versus transient landmarks and neither were the final measures of recognition, but do not report whether recognition accuracy differed across the permanent and transient landmarks at time points throughout learning (i.e. when analyzed by quarter, in keeping with the imaging results). Seeing as the permanent landmarks have a numerically greater recognition rate at final test, it would be interesting to see if this difference was significant at any time throughout learning*.

This is an excellent point, and one we were remiss not to examine. We have now conducted additional analyses which showed a significant time (quarters) by landmark type (permanent, transient) interaction (F_3,29_ = 8.045, p = 0.0005). Further examination of this result revealed the following:

Mean recognition accuracies in 1^st^ quarter

permanent landmarks = 57.7%, SD 9.7

transient landmarks = 58.1%, SD 17.7

t_31_ = -0.115, p = 0.9

Mean recognition accuracies in 2^nd^ quarter

permanent landmarks = 72.7%, SD 14.4

transient landmarks = 60.4%, SD 13.1

t_31_ = 3.517, p = 0.001

Mean recognition accuracies in 3^rd^ quarter

permanent landmarks = 80.0%, SD 15.6

transient landmarks = 70.4%, SD 16.3

t_31_ = 2.361, p = 0.02

Mean recognition accuracies in 4^th^ quarter

permanent landmarks = 79.6%, SD 18.3

transient landmarks = 77.3%, SD 14.0

t_31_ = 0.553, p = 0.6

These data are reassuring in two respects. Firstly, they show that, as expected, there was no difference in the recognition accuracy for the two landmark types at the start of the learning period. Secondly, this was also the case at the end of learning, and these 4^th^ quarter results are similar to those from the subsequent post-scan testing session. The difference lay in quarters 2 and 3, in favour of the permanent landmarks. We are not sure why this is the case, but speculate that perhaps when faced with entirely novel items, we remember them better if they are in a consistent setting.

Importantly, these findings do not materially affect our fMRI results, but rather provide additional interesting information about the nature of environmental learning. We now include these additional analyses in the Results section and consider them also in the Discussion, noting that future studies might probe this result further.

2) Regarding the principal components analysis, since the analysis separated the four measures into four distinct factors, it was not clear to me why the factor scores were used in later analyses rather than just the behavioural measures themselves. Could the authors explain the advantage of this? Furthermore, when describing the results of the PCA, the authors state that it “in particular highlighted permanence of landmarks as a distinct factor”. The loading scores don't seem to indicate that the permanence factor was any more distinct than the others. Are the authors referring to some other metric that was not included in the Results section, or do they just mean to highlight this, since permanence is the factor of interest in this study?

We used the PCA scores rather than the raw behavioural data because these provided fully orthogonalized regressors for the fMRI analysis. We now clarify this in the revised manuscript (in the subsection “fMRI: all landmark features”). We highlighted the permanence factor merely because this was the main interest for the study, but realise that this could cause confusion, so we have modified the text to make it clearer.

*3) The authors report robust and converging evidence for the RSC and POS activity relating to the permanence of landmarks, and report that the hippocampus activity relates to knowledge of location, but only in the final quarter of learning. However, given the behavioural results (i.e. that participants learned permanence/transience very well, but navigation performance was low and variable), how much can be made of the null results regarding landmark location for the other areas and learning quarters? The authors should comment on this in the Discussion*.

Thank you for highlighting this point. We have now added a comment to the Discussion noting that the null results in POS and other regions do not indicate that they contain no detailed location knowledge related to navigation performance, but that any such representations, if they do exist, are much less evident than in the hippocampus (in the subsection “Role of the hippocampus”).

*4) In a related vein, throughout learning, participants were asked about the permanence of the landmarks. While I acknowledge that this was an important aspect of the design in order to ascertain how well this information was learned, it raises the question of how much influence this orientation to permanence of the landmarks had on performance and neural activity. Did the authors debrief the participants in order to determine how much attention they were paying to learning the permanence of the landmarks? This should be addressed, and future studies could try to see if behavioural and neural responses are as robust if landmark permanence is only ascertained incidentally*.

As noted in response to point 2 of Reviewer 1 above, in the post-scan debriefing session we asked participants what they had been thinking throughout the navigation movies and how they approached learning the layout of the environment. This revealed a great deal of variety in what participants were thinking. Some tended to focus on visual characteristics of the landmarks, others paid greater attention to the overall layout of the environment’s paths or to individual landmarks’ locations along paths. Sometimes they were thinking about the landmark in view, but on other occasions were considering what came before, or what landmark might be next. Only two out of thirty two participants made reference to specifically focussing the permanence of the landmarks.

This feedback shows that participants were not overwhelmingly focused on the permanence of landmarks, even though they were aware from the initial task instructions that some landmarks would move and other would not, and that this knowledge was probed during the inter-sweep questioning. It seems participants kept their overall goal in mind, namely the requirement to learn the layout of the environment.

Additionally, in previous work ([4], [3], cited in the paper), we have demonstrated that RSC responds to landmark permanence (for everyday outdoor items) when people are engaged in a completely incidental task. Whether this is also the case when landmarks permanence is learned incidentally should be probed in future studies, as noted by the Reviewer. In the revised manuscript we mention this point and provide further details of participants’ feedback from the debriefing session (see the Introduction, Results, and the subsection “Representation of landmark permanence in RSC”).

5) Regarding the connectivity analyses, the authors only report results for the anterior hippocampus. Did no other regions show this pattern? The Methods section indicates that “any regions responsive to landmark permanence” were tested for connectivity differences. Does this mean that only the hippocampus and PHC were tested for connectivity changes? More detail about what analyses were performed should be included. Also, is the connectivity of the anterior hippocampus to RSC consistent with its anatomical connections? Is the posterior hippocampus most closely linked anatomically to RSC?

Apologies if our explanation of the connectivity analyses were not clear. There were two regions that were responsive to landmark permanence in some shape or form – RSC and POS. For the first PPI analysis it is the areas shown in Figure 4, for the second PPI analysis it is the region show in Figure 6. We used these as seed regions for a PPI connectivity analysis looking for anywhere in the whole brain that had activity with a stronger relationship to these regions while viewing permanent landmarks than transient ones. Out of the whole brain, the anterior hippocampus was the only area with which these regions showed functional connectivity related to landmark permanence.

We have now clarified the explanation of these analyses in the revised manuscript (in the subsection “Connectivity analyses”).

In terms of RSC-hippocampus connectivity, very few anatomical tracer studies have been conducted in humans. In the macaque monkey, RSC sends extensive cortical efferent projections to the medial temporal lobes (presubiculum, parasubiculum, entorhinal and parahippocampal cortex; [40]). It receives inputs from similar regions, most notably the hippocampal formation (entorhinal cortex, subiculum, presubiculum and parasubiculum), parahippocampal and perirhinal cortices. In particular, Area 29 receives a large majority of projections from the hippocampus (both subiculum and presubiculum) and entorhinal cortex (2). Thus, it is not surprising that we observe functional connectivity between RSC and areas more anteriorly in the hippocampus. This is also in keeping with many studies that have reported RSC, parahippocampal cortex and anterior hippocampus co-activing in fMRI studies (e.g. [86]). We now add further comment on this in the Discussion.

*6) The authors speculate about the role of the POS, attributing the switch from coding memorability to permanence to the subject relating landmarks to specific locations. However, since the MVB analyses did not find any evidence of POS activity relating to landmark locations, this argument seems very speculative. The authors should elaborate on their claim that, “the landmark-location representations in POS may have been discrete, while the spatial detail decoded in the hippocampus was related to the knowledge of the environment's overall layout and the location of all the permanent landmarks within it”. Is there any evidence for this? Are there other possible accounts of the patterns of POS activity? It seems that by initially tracking the most memorable landmarks, and then once permanence information is learned, switching to track this measure, the POS could be generally tuned to whatever landmarks seem the most “useful” navigationally. Before information about permanence has been ascertained, the most memorable landmarks may be the ones that are focused on, but once the participants determine which landmarks are permanent, they may shift to focus on these in order to orient and navigate better*.

We agree that the reason for the pattern of fMRI activity observed in POS is not obvious. We have now re-worked this section to make it clear that our suggestion is speculative, while also including the explanation suggested by the reviewer as an alternative possibility (in the subsection “Role of the POS”).